# Triplet character of 2D-fermion dimers arising from *s*-wave attraction via spin-orbit coupling and Zeeman splitting

Ulrich Ebling[1], Ulrich Zülicke[2] and Joachim Brand[1*]

**1** Dodd-Walls Centre for Photonic and Quantum Technologies, Centre for Theoretical Chemistry and Physics, New Zealand Institute for Advanced Study, Massey University, Private Bag 102904, North Shore, Auckland 0745, New Zealand
**2** Dodd-Walls Centre for Photonic and Quantum Technologies, School of Chemical and Physical Sciences, Victoria University of Wellington, PO Box 600, Wellington 6140, New Zealand

⋆ j.brand@massey.ac.nz

## Abstract

We theoretically study spin-1/2 fermions confined to two spatial dimensions and experiencing isotropic short-range attraction in the presence of both spin-orbit coupling and Zeeman spin splitting – a prototypical system for developing topological superfluidity in the many-body sector. Exact solutions for two-particle bound states are found to have a triplet contribution that dominates over the singlet part in an extended region of parameter space where the combined Zeeman- and center-of-mass-motion-induced spin-splitting energy is large. The triplet character of dimers is purest in the regime of weak *s*-wave interaction strength. Center-of-mass momentum is one of the parameters determining the existence of bound states, which we map out for both two- and one-dimensional types of spin-orbit coupling. Distinctive features emerging in the orbital part of the bound-state wave function, including but not limited to its *p*-wave character, provide observable signatures of unconventional pairing.

 Check for updates

# 1  Introduction

Since their early days, ultracold atomic gases have provided an intriguing avenue for exploring and simulating condensed-matter-physics phenomena. Artificial gauge fields [1–4] constitute pertinent examples for the great degree of control and ability to fine-tune parameters that are often fixed in a solid-state material. While spin-orbit coupling for quasi-free electrons in materials is fundamentally determined by the band structure [5–8] and can only be manipulated in a limited fashion via nanostructuring [9], it has recently become possible to realize synthetic versions of one-dimensional (1D) [1] and two-dimensional (2D) [10, 11] types of spin-orbit coupling for ultracold neutral atoms by means of Raman coupling with lasers. Furthermore, advances in manipulating and probing quantum gases have enabled the study of low-dimensional systems such as 2D Fermi gases [12–19] and provided detailed insights into their many-body physics via spectroscopic techniques [20]. In the future, low-dimensional systems with spin-orbit coupling and Zeeman spin splitting will allow experimentalists to create exotic condensed-matter phases such as topological superfluids that can host unconventional Majorana-fermion excitations [21]. Mean-field theory predicts the emergence of such a topological phase in $s$-wave superfluids of 2D fermions with spin-orbit coupling and large-enough Zeeman splitting [22–26].

Complementary to mean-field studies of interacting many-particle systems, analysis of the two-fermion bound state in vacuum sheds a different light on pairing that can provide crucial insight, e.g., into the strong-coupling (BEC) limit of tightly bound dimers. Dimers of fermionic atoms are further of interest in their own right, e.g., as providing qubits for quantum information processing [27]. We consider two particles (atoms) interacting only via isotropic, short range, and attractive low-energy $s$-wave scattering under the influence of synthetic spin-orbit coupling. Previous work on bound states in 3D [28–31] (see Ref. [32] for a trapped system) and in 2D [33–35] largely ignored the effects of Zeeman spin splitting. Dimers of spin-orbit-coupled fermions in a 3D gas have already been produced and probed experimentally [36].

The presence of spin-orbit coupling adds several interesting features to the two-particle problem, including the fact that strongly bound states acquire properties solely determined by the gauge field (becoming so-called "rashbons" [29]). Furthermore, Galilean invariance is broken. While total center-of-mass (COM) momentum is still conserved, it enters the bound-

state problem as a parameter. As a consequence, bound states may dissociate when scattered to large values of the COM momentum [29,35]. Also, spin-orbit coupling induces a spin-triplet component in the two-fermion bound state even with pure *s*-wave attraction [28,33,36]. This is in contrast to the situation in the absence of spin-orbit coupling, where *s*-wave attraction has no effect in the decoupled triplet channel and, thus, generates pure spin-singlet bound states (including in situations with Zeeman splitting present [36,37]). As *s*-wave interactions are usually dominant in ultra-cold atoms where higher-orbital-momentum interactions are strongly suppressed [38], spin-orbit coupling is thus a promising avenue to induce a triplet character to atomic dimers.

In this paper, we examine the exact bound-state solutions of the two-particle Schrödinger equation in 2D. By considering the effects of Zeeman spin splitting on the same footing as finite COM momentum, we extend previous works where the effects of Zeeman spin splitting were not considered in detail [33–35]. We calculate the dimer bound-state energy and delineate the critical boundary in parameter space beyond which Zeeman splitting and/or COM momentum destabilise the bound state. We also calculate, for the first time, the spin projections for the 2D bound states. The orbital part of the bound-state wave function projected onto spin-singlet and -triplet components reveals their respective *s*-wave and *p*-wave-like features. We find that both finite COM momentum and Zeeman spin-splitting favour triplet contributions to the ground state and contribute similar effects to the bound-state problem. However, while finite COM momentum favours unpolarised triplet character, the Zeeman spin splitting leads to a spin-polarised triplet character of the bound state that is associated with a chiral-*p*-wave orbital wave function. Such triplet-dominated *p*-wave dimers can be seen as a precursor of the topological superfluid that is expected to emerge in the many-body regime [21, 22, 26]. The region in parameter space where the triplet character dominates turns out to be a striking feature of systems with 2D-type (e.g., Rashba [39–41]) spin-orbit coupling, but triplet-dominated bound states are also present for 1D-type spin-orbit coupling, albeit in a reduced parameter range. The polarized spin-triplet character of bound states could be probed experimentally by spectroscopic techniques [42, 43] or spin-resolved momentum-correlation measurements on the single-particle level in the few-atom regime as recently realised in Ref. [44].

The remainder of this paper is organized as follows. In section 2, we develop the general formalism for solving the two-particle problem for any type of spin-orbit coupling and an effective Zeeman splitting that subsumes both the actual Zeeman spin splitting and finite COM momentum. In section 3, we apply this formalism to obtain bound-state solutions for 2D-type spin-orbit coupling. We first discuss the case with zero COM momentum where we obtain analytic expressions for all terms in the implicit equation for the bound-state energy as well as for the critical value of the Zeeman-spin-splitting energy above which no bound state exists. We also calculate the relative weights of spin-singlet and spin-triplet components in the two-fermion bound state and find that a parameter region exists where bound dimers have a large triplet component. We then address the case of finite COM momentum, for which no analytical results can be obtained and where the shape of the bound state is different. In section 4, we repeat our analysis for the case of 1D-type spin-orbit coupling. Compared with the 2D-type spin-orbit coupling, the parameter range for having a bound state is increased. In contrast, while dimers with a dominant triplet component still exist in this case, these occur now much closer to the threshold where the combined Zeeman- and COM-induced spin splitting destabilizes the bound state. Then, in section 5, we plot the orbital part of the bound-state wave function in relative-momentum space for the different total-spin components and parameter regimes considered in the preceding sections. We discuss possible experimental detection methods in section 6 before presenting our conclusions in section 7.

Table 1: Types of spin-orbit coupling considered in this work. Each of these is associated with a particular form of the term $\hat{\lambda}(\mathbf{p})$ in the single-particle Hamiltonian (2) and with a matrix $\mathcal{M}$ entering the transformation of momentum vectors into momentum-dependent spin splittings via Eq. (4). The constant $\lambda$, which has dimensions of velocity, quantifies the magnitude of the spin-orbit coupling.

| | 2D-Dirac | 2D-Rashba | 2D-Dresselhaus | 1D |
|---|---|---|---|---|
| $\hat{\lambda}(\mathbf{p})$ | $\lambda(p_x\,\hat{\sigma}_x + p_y\,\hat{\sigma}_y)$ | $\lambda(p_y\,\hat{\sigma}_x - p_x\,\hat{\sigma}_y)$ | $\lambda(p_x\,\hat{\sigma}_x - p_y\,\hat{\sigma}_y)$ | $\lambda\,p_x\,\hat{\sigma}_x$ |
| $\mathcal{M}$ | $\begin{pmatrix} 1 & 0 \\ 0 & 1 \\ 0 & 0 \end{pmatrix}$ | $\begin{pmatrix} 0 & -1 \\ 1 & 0 \\ 0 & 0 \end{pmatrix}$ | $\begin{pmatrix} 1 & 0 \\ 0 & -1 \\ 0 & 0 \end{pmatrix}$ | $\begin{pmatrix} 1 & 0 \\ 0 & 0 \\ 0 & 0 \end{pmatrix}$ |

## 2 General formalism for solving the two-particle problem

We consider two spin-1/2 fermions that interact via isotropic short-range interactions. Their movement is confined to the 2D plane defined by the $x$ and $y$ directions. The particles' orbital motion is coupled to their spin degree of freedom via a spin-orbit coupling that depends linearly on in-plane momentum components. In addition, Zeeman spin splitting lifts the energy degeneracy of spin projections parallel to the out-of-plane ($z$) direction. The Hamiltonian describing such a two-particle system is given by

$$\hat{H} = \hat{H}_1^{(1)} \otimes \mathbb{1} + \mathbb{1} \otimes \hat{H}_1^{(2)} + V(\mathbf{r}_1 - \mathbf{r}_2), \tag{1}$$

where $V(\mathbf{r}_1 - \mathbf{r}_2)$ is the two-particle interaction potential, and $\hat{H}_1^{(j)}$ denotes the single-particle Hamiltonian for particle $j$;

$$\hat{H}_1^{(j)} = \frac{\mathbf{p}_j^2}{2m} + h\,\hat{\sigma}_z + \hat{\lambda}(\mathbf{p}_j). \tag{2}$$

We use the symbol ˆ to indicate quantities that are operators in spin space, such as the vector of Pauli matrices $\hat{\boldsymbol{\sigma}} \equiv (\hat{\sigma}_x, \hat{\sigma}_y, \hat{\sigma}_z)$. The Zeeman-spin-splitting parameter $h$ quantifies an energy offset dependent on the $z$ component of the spin. Spin-orbit coupling is embodied in the form of $\hat{\lambda}(\mathbf{p})$. Results obtained in this work pertain to the unitarily equivalent 2D-Dirac [45], 2D-Rashba [39–41] and 2D-Dresselhaus [7,46] types of spin-orbit coupling, as well as the 1D type that is more straightforwardly realizable in cold-atom experiments [1]. Table 1 lists $\hat{\lambda}(\mathbf{p})$ for each of these four possibilities.

All of the spin-orbit-coupling types considered in this work can be expressed as

$$\hat{\lambda}(\mathbf{p}) = \lambda \sum_{a \in \{x,y,z\}} \hat{\sigma}_a \sum_{\mu \in \{x,y\}} \mathcal{M}_{a\mu}\,p_\mu \equiv \lambda\,\hat{\boldsymbol{\sigma}} \cdot \mathcal{M}\,\mathbf{p}, \tag{3}$$

with a velocity scale $\lambda$ measuring the spin-orbit-coupling strength and $\mathcal{M}_{a\mu} \in \{0, \pm 1\}$. The $3 \times 2$ matrix $\mathcal{M}$ connects spin and orbital degrees of freedom. The particular choices for $\mathcal{M}$ associated with each type of spin-orbit coupling are also specified in Table 1. In the following, a versatile theoretical treatment of different spin-orbit couplings is facilitated by introducing the 3-vector of momentum-dependent spin splittings

$$\mathbf{q} \equiv \begin{pmatrix} q_x \\ q_y \\ q_z \end{pmatrix} = \mathcal{M}\,\mathbf{p}, \tag{4}$$

such that $\hat{\lambda}(\mathbf{p}) = \lambda\,\hat{\boldsymbol{\sigma}}\cdot\mathbf{q}$. The fact that $\mathcal{M}^T\mathcal{M} = \mathbb{1}_{2\times 2}$ for all of the 2D-type spin-orbit couplings causes certain physical properties of these systems to be identical and also underpins their qualitative difference with the case of 1D-type spin-orbit coupling for which $\mathcal{M}^T\mathcal{M} = \text{diag}(1,0)$. As $\mathbf{q}$ is fully in-plane (i.e., $q_z \equiv 0$) for the spin-orbit-coupling types considered in this work, this will be implicit in the formalism. In particular, $\mathbf{q}^2 \equiv q_x^2 + q_y^2$ in all mathematical expressions below.

The eigenstates $|\alpha_j, \mathbf{p}_j\rangle$ of the single-particle Hamiltonian (2) are labelled by the individual particle's momentum $\mathbf{p}_j$ and a helicity quantum number $\alpha_j = \pm 1$ that distinguishes spin-split single-particle energy bands $\epsilon_{\alpha_j}(\mathbf{p_j})$ with dispersions

$$\epsilon_\alpha(\mathbf{p}) = \frac{\mathbf{p}^2}{2m} + \alpha\,Z(\mathbf{p})\,. \tag{5}$$

Here we introduced the effective spin-splitting energy

$$Z(\mathbf{p}) = \sqrt{h^2 + \lambda^2\,\mathbf{q}^2}\,, \tag{6}$$

that is a function of $\mathbf{p}$ via the momentum-dependent spin splitting $\mathbf{q}$ [see Eq. (4)]. Using the eigenbasis of $\hat{\sigma}_z$, the single-particle eigenspinors can be written more explicitly as

$$|\alpha, \mathbf{p}\rangle = \begin{pmatrix} e^{-i\phi/2}\,\sqrt{\frac{Z(\mathbf{p})+\alpha\,h}{2Z(\mathbf{p})}} \\ \alpha\,e^{i\phi/2}\,\sqrt{\frac{Z(\mathbf{p})-\alpha\,h}{2Z(\mathbf{p})}} \end{pmatrix}\,, \tag{7}$$

with $\phi = \arg(q_x + iq_y)$. The dispersion (5) has its minimum at the value [1]

$$\epsilon^{\min} = \begin{cases} -\frac{1}{2}\left(m\,\lambda^2 + \frac{h^2}{m\,\lambda^2}\right)\,, & \text{for } |h| \le m\,\lambda^2 \\ -|h|\,, & \text{for } |h| \ge m\,\lambda^2 \end{cases}\,. \tag{8}$$

To address the two-particle problem, we switch to COM and relative coordinates for the orbital motion,

$$\mathbf{R} = \frac{1}{2}(\mathbf{r}_1 + \mathbf{r}_2)\,, \quad \mathbf{r} = \mathbf{r}_1 - \mathbf{r}_2\,, \quad \mathbf{P} = \mathbf{p}_1 + \mathbf{p}_2\,, \quad \mathbf{p} = \frac{1}{2}(\mathbf{p}_1 - \mathbf{p}_2)\,, \tag{9}$$

and introduce the total-spin operator

$$\hat{\boldsymbol{\Sigma}} = \frac{1}{2}\left(\hat{\boldsymbol{\sigma}} \otimes \mathbb{1} + \mathbb{1} \otimes \hat{\boldsymbol{\sigma}}\right)\,, \tag{10}$$

whose eigenstates are the familiar singlet and triplet states $|SM\rangle$ with $S \in \{0,1\}$ and $M = -S, -S+1, \ldots S$ denoting eigenvalues of $\hat{\Sigma}_z$. It is possible to separate off the COM kinetic energy and write the two-particle Hamiltonian (1) in the form

$$\hat{H} = \frac{\mathbf{P}^2}{4m} + \hat{H}_\mathbf{P} + V(\mathbf{r})\,, \tag{11}$$

where $\hat{H}_\mathbf{P}$ contains the relative-motion kinetic energy and the spin-orbit coupling terms, which still depend parametrically on the COM momentum $\mathbf{P}$:

$$\hat{H}_\mathbf{P} = \frac{\mathbf{p}^2}{m} + \lambda\,\mathbf{q}\cdot\left(\hat{\boldsymbol{\sigma}} \otimes \mathbb{1} - \mathbb{1} \otimes \hat{\boldsymbol{\sigma}}\right) + 2\,\mathbf{B}_\mathbf{P}\cdot\hat{\boldsymbol{\Sigma}}\,. \tag{12}$$

Here we have introduced the abbreviation

$$\mathbf{B}_\mathbf{P} = \begin{pmatrix} \lambda\,Q_x/2 \\ \lambda\,Q_y/2 \\ h \end{pmatrix}\,, \tag{13}$$

where $\mathbf{Q} = \mathcal{M}\mathbf{P}$ with the matrix $\mathcal{M}$ associated with the spin-orbit-coupling type as per Table 1. The last term in Eq. (12) constitutes a Zeeman-splitting-like two-particle energy contribution, where $\mathbf{B_P}$ plays the role of an effective three-dimensional magnetic-field vector with in-plane components arising from the COM motion through spin-orbit coupling. However, it is not the only term that determines the spin-dependence of the two-particle energies and eigenstates. As $[\hat{\Sigma}^2, \hat{H}_\mathbf{P}] \neq 0$ due to the second term in Eq. (12), two-particle eigenstates will generally be superpositions of the eigenstates $|S\,M\rangle$ for total spin when $\lambda \neq 0$. The form of $\mathbf{B_P}$ indicates that, when spin-orbit coupling is finite, COM momentum affects the relative motion via a Zeeman-like coupling to the in-plane total-spin components. We will see below that there are certain similarities between how finite Zeeman energy $h$ and finite COM momentum $\mathbf{P}$ affect the two-particle binding energy, and how they both drive the dimer state to have predominantly triplet character when their respective Zeeman-splitting magnitudes $|h|$ and $\lambda\,|\mathbf{Q}|$ are large. However, the detailed bound-state structure is strongly influenced by the interplay of $h$ and $\lambda\,\mathbf{Q}$, such that it differs markedly in the two limits when either $\mathbf{P}$ or $h$ vanish.

We now proceed to solve the relative-motion problem embodied by the Hamiltonian $\hat{H}_\mathbf{P} + V(\mathbf{r})$. While the coordinate transformation to relative and COM coordinates of Eq. (9) has not completely removed the dependence on COM properties, it nevertheless reduces the dimensionality from four to two degrees of freedom. As the total momentum $\mathbf{P}$ is a good quantum number, the remaining $\mathbf{P}$ dependence in the relative-motion problem is solely of parametric nature as $\mathbf{P}$ can be considered to have a fixed value. The $\mathbf{P}$-dependent terms in the relative-motion Hamiltonian of Eq. (12) are proportional to the spin-orbit coupling strength $\lambda$ and thus originate directly from the spin-orbit coupling. Any of the different forms of spin-orbit coupling considered in this work (see Table 1) generate such terms. We first consider the situation of noninteracting particles and then solve the two-particle bound-state problem.

## 2.1 Case of noninteracting particles

In the absence of interactions, the two-particle eigenstates are eigenstates of $\hat{H}_\mathbf{P}$, which can be written as antisymmetrized products of the individual particles' helicity and momentum eigenstates;

$$|\mathbf{p}; \alpha_1, \alpha_2\rangle\rangle_\mathbf{P} = \mathcal{A}\big[|\mathbf{p}\rangle\,|\alpha_1, \alpha_2\rangle_{\mathbf{p},\mathbf{P}}\big] \equiv \frac{1}{\sqrt{2}}\big(|\mathbf{p}\rangle\,|\alpha_1, \alpha_2\rangle_{\mathbf{p},\mathbf{P}} - |-\mathbf{p}\rangle\,|\alpha_2, \alpha_1\rangle_{-\mathbf{p},\mathbf{P}}\big). \tag{14}$$

Here, $|\mathbf{p}\rangle$ denotes a relative-momentum eigenstate, $\alpha_j = \pm 1$, and $\mathcal{A}$ is the antisymmetrization operator. Throughout this paper, we use a notation where $|\cdot\rangle$ denotes states for the relative-orbital-motion degree of freedom, $|\cdot\rangle$ are spin states, and $|\cdot\rangle\rangle$ are full two-particle states in the product space of spin and orbital degrees of freedom. The two-particle spin state $|\alpha_1, \alpha_2\rangle_{\mathbf{p},\mathbf{P}}$ is a product state of single-particle helicity states given in Eq. (7);

$$|\alpha_1, \alpha_2\rangle_{\mathbf{p},\mathbf{P}} = |\alpha_1, \mathbf{P}/2 + \mathbf{p}\rangle \otimes |\alpha_2, \mathbf{P}/2 - \mathbf{p}\rangle. \tag{15}$$

The associated two-particle eigenenergies are

$$\varepsilon_\mathbf{P}(\alpha_1, \alpha_2, \mathbf{p}) = \frac{\mathbf{p}^2}{m} + \alpha_1 Z_+ + \alpha_2 Z_-, \tag{16}$$

with the definitions

$$Z_\pm = \sqrt{h^2 + \lambda^2\left(\frac{\mathbf{Q}}{2} \pm \mathbf{q}\right)^2}. \tag{17}$$

The relative-motion energy dispersion (16) has its minimum at

$$\epsilon_\mathbf{P}^{\min} = \begin{cases} -m\,\lambda^2 - \frac{\mathbf{B_P^2}}{m\,\lambda^2}, & \text{for } \sqrt{\mathbf{B_P^2}} \leq m\,\lambda^2 \\ -2\sqrt{\mathbf{B_P^2}}, & \text{for } \sqrt{\mathbf{B_P^2}} \geq m\,\lambda^2 \end{cases}. \tag{18}$$

For fixed $\mathbf{P}$ and $\mathbf{p}$, the states $|\alpha_1, \alpha_2\rangle_{\mathbf{p},\mathbf{P}}$ form an orthonormal basis within two-particle spin-ket space, thus providing a resolution of the unity operator $\mathbb{1}_{\mathbf{p},\mathbf{P}}$ in this subspace;

$$\sum_{\alpha_1, \alpha_2} |\alpha_1, \alpha_2\rangle_{\mathbf{p},\mathbf{P}\ \mathbf{p},\mathbf{P}}\langle \alpha_1, \alpha_2| = \mathbb{1}_{\mathbf{p},\mathbf{P}}. \tag{19}$$

## 2.2 Bound states resulting from *s*-wave attraction

The treatment of the 2D-fermion bound-state problem with short-range interactions in the absence of spin-orbit coupling is well-established [47, 48]. Recent generalizations [33–35] were developed to explore ramifications of 2D-type spin-orbit coupling. Here we extend the Green's-function formalism employed in Refs. [34, 35] to study the combined effects of spin-orbit coupling and Zeeman spin splitting.

A bound state is a solution of the Schrödinger equation

$$\left[\hat{H}_{\mathbf{P}} + V(\mathbf{r})\right]|\psi_{\mathrm{b}}\rangle\rangle = E_{\mathrm{b}}|\psi_{\mathrm{b}}\rangle\rangle, \tag{20}$$

with energy below the continuum of energies available to two unbound particles. Depending on whether or not different COM-motion states are accessible to the two-particle system under consideration, two possible threshold energies for bound states can be defined. In situations where dissociation can involve transitions between different COM momenta, stability of bound states requires their total two-particle energy $E_{\mathrm{b}} + \mathbf{P}^2/(4m)$ to be below the lowest-possible energy $2\epsilon^{\min}$ two unbound particles can have, where $\epsilon^{\min}$ is given by Eq. (8). In this case, one should be looking for eigenstates $|\psi_{\mathrm{b}}\rangle\rangle$ of $\hat{H}_{\mathbf{P}} + V(\mathbf{r})$ that satisfy $E_{\mathrm{b}} < E_{\mathrm{th}}^{\mathrm{abs}}$ with

$$E_{\mathrm{th}}^{\mathrm{abs}} = 2\,\epsilon^{\min} - \frac{\mathbf{P}^2}{4m} \equiv \begin{cases} -m\,\lambda^2 - \frac{h^2}{m\,\lambda^2} - \frac{\mathbf{P}^2}{4m}, & \text{for } |h| \leq m\,\lambda^2 \\ -2\,|h| - \frac{\mathbf{P}^2}{4m}, & \text{for } |h| \geq m\,\lambda^2 \end{cases}. \tag{21}$$

Alternatively, if the COM motion of the two-particle system is considered to be conserved, we can focus only on the relative-motion dynamics for a two-particle system with fixed COM momentum $\mathbf{P}$. Then the threshold energy for bound states is given by the minimum energy $\epsilon_{\mathbf{P}}^{\min}$ available to the relative motion of two unbound particles at the fixed COM momentum $\mathbf{P}$ [see Eq. (18)], i.e., the bound states need to satisfy $E_{\mathrm{b}} < E_{\mathrm{th}}^{\mathrm{rel}}$ with

$$E_{\mathrm{th}}^{\mathrm{rel}} = \epsilon_{\mathbf{P}}^{\min} \equiv \begin{cases} -m\,\lambda^2 - \frac{h^2}{m\,\lambda^2} - \frac{\mathbf{Q}^2}{4m}, & \text{for } \sqrt{h^2 + \frac{\lambda^2 \mathbf{Q}^2}{4}} \leq m\,\lambda^2 \\ -2\sqrt{h^2 + \frac{\lambda^2 \mathbf{Q}^2}{4}}, & \text{for } \sqrt{h^2 + \frac{\lambda^2 \mathbf{Q}^2}{4}} \geq m\,\lambda^2 \end{cases}. \tag{22}$$

All three energies $E_{\mathrm{b}}$, $E_{\mathrm{th}}^{\mathrm{abs}}$ and $E_{\mathrm{th}}^{\mathrm{rel}}$ depend parametrically on the COM momentum $\mathbf{P}$, and $E_{\mathrm{th}}^{\mathrm{abs}} = E_{\mathrm{th}}^{\mathrm{rel}}$ for $\mathbf{P} = 0$. For 2D-type spin-orbit coupling and $\mathbf{B}_{\mathbf{P}}^2 \leq m^2\lambda^4$, $E_{\mathrm{th}}^{\mathrm{abs}}$ and $E_{\mathrm{th}}^{\mathrm{rel}}$ are identical even when $\mathbf{P} \neq 0$, but $E_{\mathrm{th}}^{\mathrm{abs}} < E_{\mathrm{th}}^{\mathrm{rel}}$ when $\mathbf{P} \neq 0$ for 1D-type spin-orbit coupling and/or $\mathbf{B}_{\mathbf{P}}^2 \geq m^2\lambda^4$. In the present work, we adopt $E_{\mathrm{th}}^{\mathrm{rel}}$ as the threshold energy to determine the existence of two-particle bound states and to calculate their binding energy

$$\epsilon_{\mathrm{b}} \equiv E_{\mathrm{th}}^{\mathrm{rel}} - E_{\mathrm{b}}. \tag{23}$$

Such states are only metastable when $E_{\mathrm{th}}^{\mathrm{abs}} \leq E_{\mathrm{b}} < E_{\mathrm{th}}^{\mathrm{rel}}$, but they can still be sufficiently long-lived, and therefore accessible experimentally, in situations when COM-changing processes are weak. Note that both threshold energies are negative, $E_{\mathrm{th}}^{\mathrm{abs}} \leq E_{\mathrm{th}}^{\mathrm{rel}} \leq 0$, and thus $E_{\mathrm{b}} < 0$. At the same time, the binding energy is defined to be positive; $\epsilon_{\mathrm{b}} > 0$.

The Schrödinger equation (20) can be formally solved via

$$|\psi_{\mathrm{b}}\rangle\rangle = \frac{1}{E_{\mathrm{b}} - \hat{H}_{\mathbf{P}}} V |\psi_{\mathrm{b}}\rangle\rangle, \tag{24}$$

as the denominator on the right-hand side is never zero because the bound-state energy is outside the eigenvalue spectrum of $\hat{H}_{\mathbf{P}}$. We can expand the full bound-state wave function with respect to the relative-momentum eigenbasis,

$$|\psi_{\mathrm{b}}\rangle\rangle = \int \frac{d^2 p'}{(2\pi\hbar)^2} \, (\mathbf{p}'|\psi_{\mathrm{b}}\rangle\rangle \, |\mathbf{p}'\rangle, \tag{25}$$

keeping in mind that the expansion "coefficients" $(\mathbf{p}'|\psi_{\mathrm{b}}\rangle\rangle \equiv |\psi_{\mathrm{b}}(\mathbf{p}')\rangle$ are actually still kets in spin space that parametrically depend on the total momentum $\mathbf{P}$ and are nonorthogonal for different arguments $\mathbf{p}'$. Inserting the expansion (25) on the r.h.s. of Eq. (24) and projecting both sides onto $(\mathbf{p}|$, we obtain

$$|\psi_{\mathrm{b}}(\mathbf{p})\rangle = \frac{1}{E_{\mathrm{b}} - \hat{H}_{\mathbf{P}}} \int \frac{d^2 p'}{(2\pi\hbar)^2} \, (\mathbf{p}|V|\mathbf{p}') \, |\psi_{\mathrm{b}}(\mathbf{p}')\rangle \equiv \hat{G}_{\mathbf{P}}(E_{\mathrm{b}}, \mathbf{p}) \int \frac{d^2 p'}{(2\pi\hbar)^2} \, (\mathbf{p}|V|\mathbf{p}') \, |\psi_{\mathrm{b}}(\mathbf{p}')\rangle, \tag{26}$$

where the Green's function $\hat{G}_{\mathbf{P}}(E, \mathbf{p})$ is an operator (a $4 \times 4$ matrix) in two-particle spin space.

A general isotropic interaction potential can be expanded in partial waves, yielding

$$(\mathbf{p}|V|\mathbf{p}') = \sum_{l=-\infty}^{\infty} V_l(\mathbf{p}, \mathbf{p}') \, e^{il(\phi_{\mathbf{p}} - \phi_{\mathbf{p}'})}. \tag{27}$$

Here $\phi_{\mathbf{p}}$ is the polar angle of the vector $\mathbf{p}$. Furthermore, as the eigenstates $|S\,M\rangle$ of total spin form a basis in two-particle spin space, we can expand

$$|\psi_{\mathrm{b}}(\mathbf{p})\rangle = \sum_{S,M} |S\,M\rangle\langle S\,M|\psi_{\mathrm{b}}(\mathbf{p})\rangle, \tag{28}$$

where, due to the antisymmetry requirement of two-fermion wave functions, $\langle 0\,0|\psi_{\mathrm{b}}(\mathbf{p})\rangle$ must be an even function of $\mathbf{p}$, whereas the functions $\langle 1\,M|\psi_{\mathrm{b}}(\mathbf{p})\rangle$ must be odd. Inserting both (27) and (28) into the r.h.s. of (26) yields

$$|\psi_{\mathrm{b}}(\mathbf{p})\rangle = \sum_{S,M} \hat{G}_{\mathbf{P}}(E_{\mathrm{b}}, \mathbf{p}) |S\,M\rangle \sum_l \int \frac{d^2 p'}{(2\pi\hbar)^2} \, V_l(\mathbf{p}, \mathbf{p}') \, e^{il(\phi_{\mathbf{p}} - \phi_{\mathbf{p}'})} \, \langle S\,M|\psi_{\mathrm{b}}(\mathbf{p}')\rangle. \tag{29}$$

For the case of short-range, low-energy, $s$-wave scattering, $V_l(\mathbf{p}, \mathbf{p}') \to V_0 \, \delta_{l,0}$ and the integral on the r.h.s. of Eq. (29) remains finite (vanishes) for the singlet(triplet)-state contribution(s) because the integrand is an even (odd) function of $\mathbf{p}'$. Thus Eq. (29) simplifies to

$$|\psi_{\mathrm{b}}(\mathbf{p})\rangle = \hat{G}_{\mathbf{P}}(E_{\mathrm{b}}, \mathbf{p}) |0\,0\rangle \, V_0 \int \frac{d^2 p'}{(2\pi\hbar)^2} \, \langle 0\,0|\psi_{\mathrm{b}}(\mathbf{p}')\rangle, \tag{30}$$

and it follows that the bound-state wave function is obtained by the action of the Green's function on the singlet state;

$$|\psi_{\mathrm{b}}(\mathbf{p})\rangle = N_{\mathbf{P}} \, \hat{G}_{\mathbf{P}}(E_{\mathrm{b}}, \mathbf{p}) |0\,0\rangle. \tag{31}$$

The modulus of the $\mathbf{P}$-dependent normalization factor $N_{\mathbf{P}}$ is determined by the normalization condition for $|\psi_{\mathrm{b}}\rangle\rangle$;

$$\langle\langle\psi_{\mathrm{b}}|\psi_{\mathrm{b}}\rangle\rangle = \int \frac{d^2 p}{(2\pi\hbar)^2} \int \frac{d^2 p'}{(2\pi\hbar)^2} \, (\mathbf{p}|\mathbf{p}') \, \langle\psi_{\mathrm{b}}(\mathbf{p})|\psi_{\mathrm{b}}(\mathbf{p}')\rangle = \int \frac{d^2 p}{(2\pi\hbar)^2} \, \langle\psi_{\mathrm{b}}(\mathbf{p})|\psi_{\mathrm{b}}(\mathbf{p})\rangle = 1, \tag{32}$$

where we used the orthogonality relation $(\mathbf{p}|\mathbf{p}') = 2\pi\hbar^2 \, \delta(\mathbf{p}' - \mathbf{p})$ for relative-momentum eigenstates. The right-most equality from Eq. (32) demonstrates that a spin-space ket $|\psi_{\mathrm{b}}(\mathbf{p})\rangle$

is not itself normalized to unity. Inserting (31) and recognizing also that $\hat{G}_{\mathbf{P}}(E_b, \mathbf{p})$ is a Hermitian operator in two-particle spin space yields

$$|N_{\mathbf{P}}| = \left\{ \int \frac{d^2 p}{(2\pi\hbar)^2} \, \langle 0\,0| [\hat{G}_{\mathbf{P}}(E_b, \mathbf{p})]^2 |0\,0\rangle \right\}^{-\frac{1}{2}}. \tag{33}$$

The amplitudes $\langle S\,M|\psi_b(\mathbf{p})\rangle$ for the $s$-wave-attraction-generated bound state (31) can be neatly expressed in terms of matrix elements of the Green's function,

$$\langle S\,M|\psi_b(\mathbf{p})\rangle = N_{\mathbf{P}} \, \langle S\,M|\hat{G}_{\mathbf{P}}(E_b, \mathbf{p})|0\,0\rangle, \tag{34}$$

for which we have obtained the general analytical expressions (see Appendix B for details of the derivation)

$$\langle 0\,0|\hat{G}_{\mathbf{P}}(E_b, \mathbf{p})|0\,0\rangle = -\frac{s}{d}\left(s^2 - 4h^2 - \lambda^2\mathbf{Q}^2\right) \equiv -\frac{s}{d}\left(s^2 - 4\mathbf{B}_{\mathbf{P}}^2\right), \tag{35a}$$

$$\langle 1\,0|\hat{G}_{\mathbf{P}}(E_b, \mathbf{p})|0\,0\rangle = -\frac{2\lambda}{d}\left[2\lambda h\,\mathbf{Q}\cdot\mathbf{q} - i\,s\,(\mathbf{Q}\times\mathbf{q})_z\right], \tag{35b}$$

$$\langle 1\,1|\hat{G}_{\mathbf{P}}(E_b, \mathbf{p})|0\,0\rangle = -\frac{\sqrt{2}\,\lambda}{d}\left[\lambda^2\mathbf{Q}\cdot\mathbf{q}\,(Q_x - i\,Q_y) + (s^2 + 2s\,h)(q_x - i\,q_y)\right], \tag{35c}$$

$$\langle 1\,{-}1|\hat{G}_{\mathbf{P}}(E_b, \mathbf{p})|0\,0\rangle = -\frac{\sqrt{2}\,\lambda}{d}\left[\lambda^2\mathbf{Q}\cdot\mathbf{q}\,(Q_x + i\,Q_y) + (s^2 - 2s\,h)(q_x + i\,q_y)\right]. \tag{35d}$$

Here we used the abbreviation

$$d = s^4 - 4s^2\left(\lambda^2\mathbf{q}^2 + h^2 + \lambda^2\mathbf{Q}^2/4\right) + 4\lambda^4(\mathbf{Q}\cdot\mathbf{q})^2 \equiv s^2\left(s^2 - 4\mathbf{B}_{\mathbf{P}}^2 - 4\lambda^2\mathbf{q}^2\right) + 4\lambda^4(\mathbf{Q}\cdot\mathbf{q})^2, \tag{36}$$

and $s = \mathbf{p}^2/m - E_b$. The expressions (34) to (36) for the bound-state wave function extend similar expressions given in Ref. [35] for the case with $h = 0$ by fully accounting for nonzero Zeeman spin splitting. Based on the expansion (28) with amplitudes (34), we define the fractional weights of total-spin eigenstates in the bound-state wave function as

$$N_{SM} = \frac{\int d^2 p \, \left|\langle S\,M|\hat{G}_{\mathbf{P}}(E_b, \mathbf{p})|0\,0\rangle\right|^2}{\sum_{S,M} \int d^2 p \, \left|\langle S\,M|\hat{G}_{\mathbf{P}}(E_b, \mathbf{p})|0\,0\rangle\right|^2}. \tag{37}$$

## 2.3 Binding energy from the Bethe-Peierls boundary condition

The characteristic equation for the bound-state energy can be found by projecting Eq. (30) onto the singlet state and integrating over momentum, which yields

$$\frac{1}{V_0} = \int \frac{d^2 p}{(2\pi\hbar)^2} \, \langle 0\,0|\hat{G}_{\mathbf{P}}(E_b, \mathbf{p})|0\,0\rangle, \tag{38}$$

with the matrix element of the Green's function between singlets given explicitly in Eq. (35a). Although principally correct, Eq. (38) turns out to be impractical for determining bound-state energies because the integral on its r.h.s. is ultraviolet-divergent. *Ad hoc* cut-offs have sometimes been introduced to circumvent this issue [35]. Here we address the problem using the Bethe-Peierls boundary condition for a scattering wave function in 2D.

We consider the equivalent of Eq. (31) in real space,

$$|\psi_b(\mathbf{r})\rangle = N_{\mathbf{P}} \, \hat{g}_{\mathbf{P}}(E_b, \mathbf{r})|0\,0\rangle, \tag{39}$$

where $g(E_{\mathrm{b}}, \mathbf{r})$ denotes the real-space Green's function[1]

$$\hat{g}_{\mathbf{P}}(E_{\mathrm{b}}, \mathbf{r}) = \int \frac{d^2 p}{(2\pi\hbar)^2} \, e^{\frac{i\mathbf{p}\cdot\mathbf{r}}{\hbar}} \, \hat{G}_{\mathbf{P}}(E_{\mathrm{b}}, \mathbf{p}). \tag{40}$$

Employing the resolution of unity Eq. (19) in two-particle spin space in terms of eigenstates of $\hat{H}_{\mathbf{P}}$, the real-space Green's function's matrix element between singlets is found as

$$\langle 0\,0|\hat{g}_{\mathbf{P}}(E_{\mathrm{b}}, \mathbf{r})|0\,0\rangle =$$
$$\frac{1}{(2\pi\hbar)^2} \int d^2 p \sum_{\alpha_1,\alpha_2} e^{\frac{i\mathbf{p}\cdot\mathbf{r}}{\hbar}} \, |\langle 00|\alpha_1, \alpha_2\rangle_{\mathbf{p},\mathbf{P}}|^2 \left( \frac{1}{E_{\mathrm{b}} - \varepsilon_{\mathbf{P}}(\alpha_1, \alpha_2, \mathbf{p})} - \frac{1}{E_{\mathrm{b}} - \mathbf{p}^2/m} \right)$$
$$+ \frac{1}{(2\pi\hbar)^2} \int d^2 p \, e^{\frac{i\mathbf{p}\cdot\mathbf{r}}{\hbar}} \, \frac{1}{E_{\mathrm{b}} - \mathbf{p}^2/m}, \tag{41}$$

where the first term on the right-hand side is regular and we have separated off the second term, which diverges logarithmically for $|\mathbf{r}| = 0$. Noting that $E_{\mathrm{b}} < 0$, this second term evaluates explicitly to a modified Bessel function

$$\frac{1}{(2\pi\hbar)^2} \int d^2 p \, e^{\frac{i\mathbf{p}\cdot\mathbf{r}}{\hbar}} \, \frac{1}{E_{\mathrm{b}} - \mathbf{p}^2/m} = -\frac{m}{2\pi\hbar^2} K_0(|\mathbf{r}| \sqrt{-m E_{\mathrm{b}}}/\hbar), \tag{42}$$

for which the small-argument behavior is known: $K_0(\xi) = -\gamma - \ln(\xi/2) + o(\xi)$. Thus we can expand the Green's function in the short-range limit $|\mathbf{r}| \to 0$ as

$$\langle 0\,0|\hat{g}_{\mathbf{P}}(E_{\mathrm{b}}, \mathbf{r})|0\,0\rangle = \frac{m}{2\pi\hbar^2} \left[ \ln\left( |\mathbf{r}| \sqrt{-m E_{\mathrm{b}}}/2\hbar \right) + \gamma + F_{\mathbf{P}}(E_{\mathrm{b}}, h) + o(|\mathbf{r}|) \right], \tag{43}$$

with finite spin-orbit coupling giving rise to the $\mathbf{r}$-independent contribution

$$F_{\mathbf{P}}(E_{\mathrm{b}}, h) = \frac{1}{m} \int \frac{d^2 p}{2\pi} \sum_{\alpha_1,\alpha_2} |\langle 0\,0|\alpha_1, \alpha_2\rangle_{\mathbf{p},\mathbf{P}}|^2 \left( \frac{1}{E_{\mathrm{b}} - \varepsilon_{\mathbf{P}}(\alpha_1, \alpha_2, \mathbf{p})} - \frac{1}{E_{\mathrm{b}} - \mathbf{p}^2/m} \right)$$
$$= \frac{1}{m} \int \frac{d^2 p}{2\pi} \left( \langle 0\,0|\hat{G}_{\mathbf{P}}(E_{\mathrm{b}}, \mathbf{p})|0\,0\rangle + \frac{1}{s} \right) \equiv \frac{4\lambda^2}{m} \int \frac{d^2 p}{2\pi} \left( \frac{\lambda^2 (\mathbf{Q}\cdot\mathbf{q})^2 - s^2 \mathbf{q}^2}{s \, d} \right). \tag{44}$$

Here we made use of Eq. (35a) to derive the last equality in Eq. (44).

For the singlet component of the bound-state wave function, the short-range behavior is described by the Bethe-Peierls boundary conditions [48] which, for 2D systems, have a logarithmic divergence

$$\langle 0\,0|\psi_{\mathrm{b}}(\mathbf{r})\rangle = N_{\mathbf{P}} \langle 0\,0|\hat{g}_{\mathbf{P}}(E_{\mathrm{b}}, \mathbf{r})|0\,0\rangle \propto \ln(|\mathbf{r}|/a_{2D}) + o(|\mathbf{r}|). \tag{45}$$

Unlike their 3D counterparts, in two spatial dimensions there is no additional term added due to the spin-orbit coupling [49]. Matching the Bethe-Peierls boundary condition (45) to the short-range limit of the singlet projection for the bound-state wave function given in Eq. (43), we obtain the implicit equation

$$\gamma + \ln\left( a_{2D} \sqrt{-m E_{\mathrm{b}}}/2\hbar \right) + F_{\mathbf{P}}(E_{\mathrm{b}}, h) = 0 \tag{46}$$

for the bound state energy $E_{\mathrm{b}}$. For convenience, we parameterize the two-particle interaction strength in terms of the energy scale $\epsilon_0 = \hbar^2/m a_{2D}^2$. Introducing characteristic units in terms

---

[1]Note that $\hat{g}_{\mathbf{P}}(E_{\mathrm{b}}, \mathbf{r})$ pertains to real space only for the two particles' relative motion but is still in momentum space at constant $\mathbf{P}$ for their COM motion.

of the spin-orbit-coupling strength $\lambda$ allows us to define the dimensionless quantities

$$\tilde{E}_b = \frac{E_b}{m\lambda^2}, \tag{47a}$$

$$\tilde{h} = \frac{h}{m\lambda^2}, \tag{47b}$$

$$\tilde{\epsilon}_0 = \frac{\epsilon_0}{m\lambda^2}. \tag{47c}$$

In terms of these, Eq. (46) becomes

$$\gamma + \ln\left(\frac{1}{2}\sqrt{\frac{-\tilde{E}_b}{\tilde{\epsilon}_0}}\right) = -F_{\mathbf{P}}(\tilde{E}_b, \tilde{h}). \tag{48}$$

In the absence of spin-orbit coupling, i.e., for $\lambda \to 0$, the r.h.s of Eq. (48) vanishes and $E_b \to -4e^{-2\gamma}\epsilon_0$ is obtained, reproducing the well-known result [48] for the two-particle bound-state energy in two spatial dimensions.

# 3 Bound-state properties for 2D-type spin-orbit coupling

In this section, we consider the bound-state problem for the case of 2D-type spin-orbit couplings of the Dirac, Rashba or Dresselhaus forms (see Table 1). The bound state's energy and conditions for its existence are the same for all three forms because they give rise to the same $F_{\mathbf{P}}(E_b, h)$ and $E_{\text{th}}^{\text{rel}}$. This is a direct consequence of the relation $\mathcal{M}^T\mathcal{M} = \mathbb{1}_{2\times2}$ holding for the three 2D-type spin-orbit couplings, which ensures the universal forms $\mathbf{q}^2 \equiv \mathbf{p}^2$, $\mathbf{Q}^2 \equiv \mathbf{P}^2$ and $\mathbf{Q} \cdot \mathbf{q} \equiv \mathbf{P} \cdot \mathbf{p}$ for momentum-dependent terms in the expressions (44) and (22). For the same reason, the singlet component of the bound-state wave function is also the same for all 2D-type spin-orbit couplings, but this universality does not extend to the triplet components as these are sensitive to the particular form of $\mathcal{M}$ [see Eqs. (35)].

## 3.1 Case of zero center-of-mass momentum

We first assume $\mathbf{P} = \mathbf{0}$. In this case, we can obtain an analytic expression for the quantity $F_{\mathbf{P}}(\tilde{E}_b, \tilde{h})$ that appears on the r.h.s. of Eq. (48);

$$F_0(\tilde{E}_b, \tilde{h}) = \frac{1}{4(\tilde{h}^2 + \tilde{E}_b)}\left[-\tilde{E}_b\ln\left(\frac{\tilde{E}_b^2}{\tilde{E}_b^2 - 4\tilde{h}^2}\right)\right.$$

$$\left.-\begin{cases} \frac{2(2\tilde{h}^2+\tilde{E}_b)}{\sqrt{-1-\tilde{h}^2-\tilde{E}_b}}\left[\frac{\pi}{2} - \arctan\left(\frac{-\tilde{E}_b-2}{2\sqrt{-1-\tilde{h}^2-\tilde{E}_b}}\right)\right]\right], & \text{for } \tilde{E}_b \leq -1-\tilde{h}^2 \\ \frac{2(2\tilde{h}^2+\tilde{E}_b)}{\sqrt{1+\tilde{h}^2+\tilde{E}_b}}\operatorname{arcoth}\left(\frac{-\tilde{E}_b-2}{2\sqrt{1+\tilde{h}^2+\tilde{E}_b}}\right)\right], & \text{for } -1-\tilde{h}^2 < \tilde{E}_b < -2|\tilde{h}| \end{cases}. \tag{49}$$

For given dimensionless interaction strength $\tilde{\epsilon}_0$ and Zeeman energy $\tilde{h}$, Eq. (48) constitutes an implicit equation for the dimensionless bound-state energy $\tilde{E}_b$, which needs to be below the threshold $\tilde{E}_{\text{th}} \equiv E_{\text{th}}^{\text{rel}}/(m\lambda^2)$ with $E_{\text{th}}^{\text{rel}}$ from Eq. (22). We find that there is at most one such solution of Eq. (48) anywhere in parameter space. Figure 1 shows plots of the dimensionless binding energy $\tilde{\epsilon}_b = \epsilon_b/(m\lambda^2)$, with $\epsilon_b$ defined in Eq. (23), as a function of $\tilde{\epsilon}_0$ and $\tilde{h}$.

The nature of the bound-state solutions depends on the value of $\tilde{h}$. For $|\tilde{h}| \leq 1$ we have $\tilde{E}_{\text{th}} = -1-\tilde{h}^2$ according to Eq. (22). Since $\tilde{E}_b < \tilde{E}_{\text{th}}$, the upper option for the last term in the

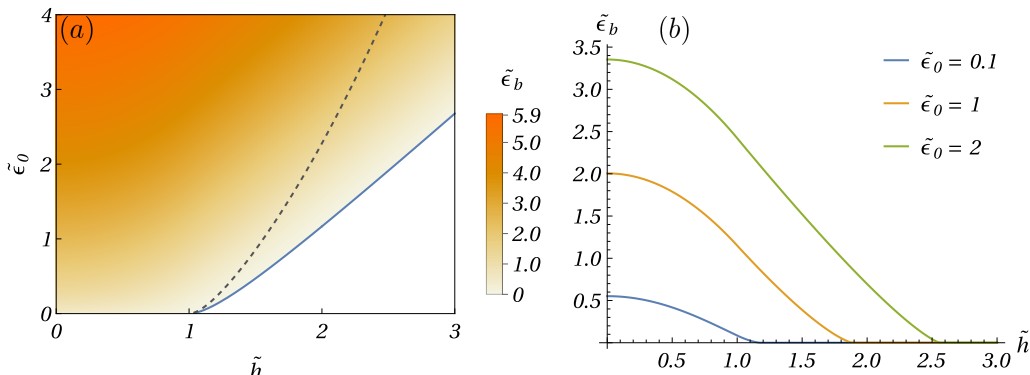

Figure 1: Binding energy of a 2D-fermion dimer with zero center-of-mass momentum ($\mathbf{P} = \mathbf{0}$) formed in the presence of 2D-type spin-orbit coupling and Zeeman splitting. Orange color in panel (a) indicates the parameter region where a bound state exists. The analytical result (51) for its boundary is indicated by the solid blue curve, and the dashed black curve indicates the dividing line between regions where the two different forms of $F_0(\tilde{E}_b, \tilde{h})$ given in Eq. (49) apply. Here $\tilde{\epsilon}_0 \equiv \epsilon_0/(m\lambda^2)$ and $\tilde{h} \equiv h/(m\lambda^2)$ are the $s$-wave interaction strength and the Zeeman energy, respectively, measured in units of the spin-orbit-coupling energy scale $m\lambda^2$. The dimensionless binding energy $\tilde{\epsilon}_b \equiv \epsilon_b/(m\lambda^2)$ is represented by the color scale in panel (a) and plotted as a function of $\tilde{h}$ for selected values of $\tilde{\epsilon}_0$ in panel (b).

expression (49) for $F_0(\tilde{E}_b, \tilde{h})$ applies in this case. In particular, solving Eq. (48) with $F_0(\tilde{E}_b, 0)$ obtained from the $|h| \to 0$ limit of Eq. (49) yields the result given in Ref. [33] for the binding energy in the absence of Zeeman spin splitting.

In contrast, for $|\tilde{h}| > 1$ we have $\tilde{E}_{\text{th}} = -2|\tilde{h}|$, and the region where a bound state exists has two parts. The two parts are distinguished by whether $\tilde{E}_b \le -1 - \tilde{h}^2$ or $-1 - \tilde{h}^2 < \tilde{E}_b < -2|\tilde{h}|$ is satisfied and, accordingly, which form of the last term in (49) is applicable. The boundary dividing these two regions can be found by letting $\tilde{E}_b \to -1 - \tilde{h}^2$ in Eq (48) with $F_0(\tilde{E}_b, \tilde{h})$ from Eq. (49) on the r.h.s., which yields (see Appendix C for mathematical details)

$$\tilde{\epsilon}_0^{\text{div}}(\tilde{h}) = e^{2\gamma + 2} \frac{1 + \tilde{h}^2}{4} \left( \frac{\tilde{h}^2 - 1}{\tilde{h}^2 + 1} \right)^{1 + \tilde{h}^2} \Theta(|\tilde{h}| - 1). \tag{50}$$

Here $\Theta(\cdot)$ denotes the Heaviside step function. In Fig. 1, we plot $\tilde{\epsilon}_0^{\text{div}}(\tilde{h})$ calculated according to Eq. (50) as the dashed black curve.

While a bound state always exists for small-enough Zeeman splitting $|\tilde{h}| \le 1$, having a bound state for $|\tilde{h}| > 1$ requires sufficiently strong attractive interactions. The white area shown in Fig. 1(a) indicates the parameter range for which no bound state exists. The minimum dimensionless interaction strength $\tilde{\epsilon}_0^{\text{crit}}(\tilde{h})$ needed to maintain a bound state at finite Zeeman splitting is obtained by substituting the threshold energy $\tilde{E}_{\text{th}} = -2|\tilde{h}|$ applicable for $|\tilde{h}| > 1$ into Eq (48), yielding (details of the derivation are provided in Appendix C)

$$\tilde{\epsilon}_0^{\text{crit}}(\tilde{h}) = e^{2\gamma} \frac{|\tilde{h}|}{2} \left( 2 \frac{|\tilde{h}| - 1}{|\tilde{h}|} \right)^{\frac{2}{2 - |\tilde{h}|}} \Theta(|\tilde{h}| - 1). \tag{51}$$

We show $\tilde{\epsilon}_0^{\text{crit}}(\tilde{h})$ as the solid blue line in Fig. 1(a). As is apparent from Fig. 1(b), the binding energy approaches zero continuously at this boundary. In the Zeeman-splitting-dominated limit $|\tilde{h}| \gg 1$, $\tilde{\epsilon}_0^{\text{crit}}(\tilde{h}) \to e^{2\gamma} \tilde{h}/2$, which is reminiscent of the Chandrasekhar-Clogston criterion [50, 51] for the stability of $s$-wave pairing against spin paramagnetism. For a two-particle problem without spin-orbit coupling, the critical interaction strength $\epsilon_0^{\text{crit}}(h) = e^{2\gamma} |h|/2$ emerges

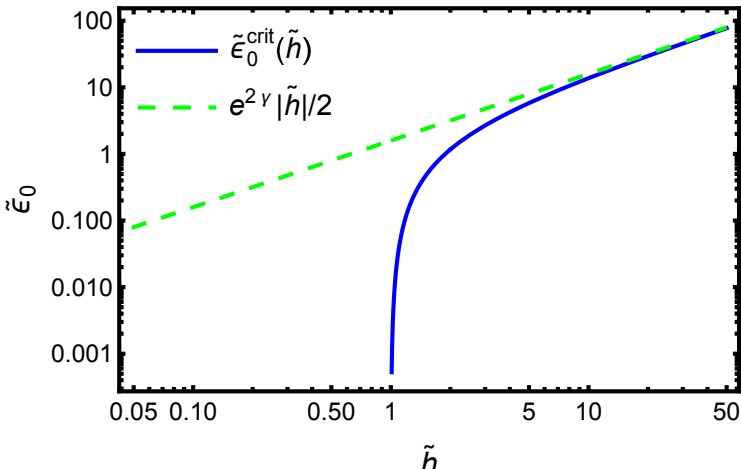

Figure 2: Boundary of the region in $\tilde{\epsilon}_0$-$\tilde{h}$ space with a two-fermion bound state. The solid blue curve plots Eq. (51) for the dimensionless critical interaction strength $\tilde{\epsilon}_0^{\text{crit}}(\tilde{h})$ above which a bound state exists in the presence of spin-orbit coupling. Its asymptote $e^{2\gamma}|\tilde{h}|/2$ for large dimensionless Zeeman coupling $\tilde{h} \equiv h/(m\lambda^2)$ is indicated by the dashed green line. For pairs of parameter values $(\epsilon_0, h)$ from the region above this line, bound states are formed also in the absence of spin-orbit coupling. In contrast, in the region between the two curves, *s*-wave-attraction-generated bound states would not exist without spin-orbit coupling.

from equating the two-fermion binding energy [48] $4e^{-2\gamma}\epsilon_0$ with the Zeeman-splitting energy $2|h|$. Figure 2 illustrates how spin-orbit coupling enlarges the region in parameter space where two-fermion binding due to *s*-wave attraction is possible in the presence of Zeeman spin splitting.

It is known that the combination of *s*-wave attraction and spin-orbit coupling can result in behavior analogous to a system subject to *p*-wave interactions without spin-orbit coupling [22]. In our case, this would manifest as having also triplet components of the bound-state wave function, even though the attractive potential is *s*-wave. In the case without spin-orbit coupling, *s*-wave interactions at low energy only lead to binding in the singlet channel, and the overlap of the wave function to the triplet component would vanish, as the Green's function in Eq. (31) would be diagonal in spin space. Turning on a Zeeman splitting $h > 0$ in the absence of spin-orbit coupling does not affect the singlet dimer until its complete destabilization when the energy $-2h$ of the triplet state $|1-1\rangle$ goes below the bound-state energy. However, with spin-orbit coupling present, the *s*-wave interaction potential still projects the wave function onto the singlet but the subsequent free propagation in the presence of spin-orbit coupling rotates parts of the wave function back into the triplet channel. Here we are interested to understand in which regime a large triplet component of the bound state develops. As the triplet state $|1-1\rangle$ is energetically favored for large Zeeman splitting, it can be expected to dominate the system. The same behavior is also seen in the BCS mean-field theory of the many-body system where the topological superfluid with *p*-wave order parameter emerges for large Zeeman splitting [21, 22, 26]. As we see below, the bound-state wave function is indeed dominated by the triplet components in an extended region near the critical Zeeman splitting $\tilde{h}^{\text{crit}}(\tilde{\epsilon}_0)$ obtained by inverting the expression for $\tilde{\epsilon}_0^{\text{crit}}(\tilde{h})$ from Eq. (51).

We obtain the fractional weights of total-spin eigenstates contributing to the bound-state wave function via numerical evaluation of Eq. (37), utilizing the expressions (35) for Green's-function matrix elements. As the r.h.s. of Eq. (35b) vanishes identically for $\mathbf{P} = \mathbf{0}$, the triplet state $|10\rangle$ makes no contribution to the bound state in the presently considered case. Fig-

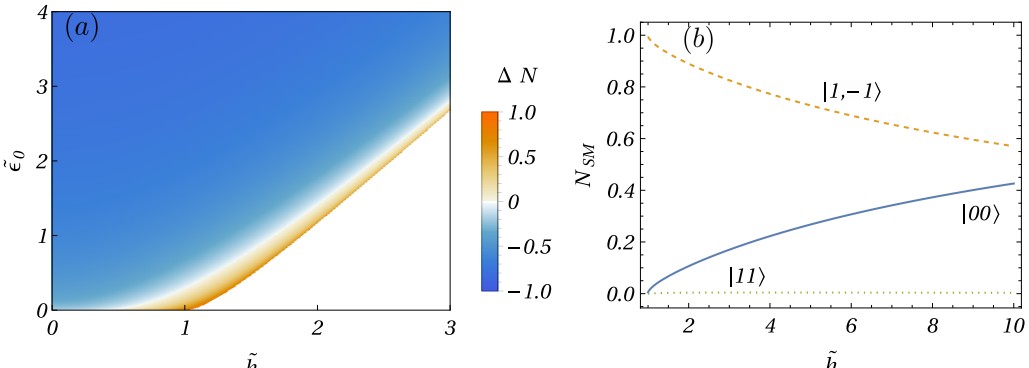

Figure 3: Triplet-state admixture to 2D-fermion bound states with zero center-of-mass momentum. The color scale in panel (a) visualizes the difference $\Delta N = \sum_M N_{1M} - N_{00}$ between the combined fractional weights of triplet states contributing to the dimer and that of the singlet-state contribution. Panel (b) shows plots of $N_{SM}$ (except for $N_{10} = 0$) for total-spin eigenstates in bound states formed for parameter combinations $(\tilde{h}, \tilde{\epsilon}_0) \equiv (\tilde{h}^{\mathrm{crit}}, \tilde{\epsilon}_0^{\mathrm{crit}})$ corresponding to the solid blue line shown in Fig. 1(a), with $\tilde{\epsilon}_0^{\mathrm{crit}}$ [$\tilde{h}^{\mathrm{crit}}$] given by Eq. (51) [by inverting Eq. (51)].

ure 3(a) shows the difference $\Delta N \equiv \sum_M N_{1M} - N_{00}$ between the combined fractional weights for triplet states and that of the singlet state within the region of parameter space depicted in Fig. 1(a). The quantity $\Delta N$ constitutes a measure for the triplet character of the two-particle bound state as, by construction, $-1 \leq \Delta N \leq 1$, where $\Delta N = 1$ indicates a pure triplet and $\Delta N = -1$ a pure singlet state. From Fig. 3(a), we see that the triplet contribution to the bound state dwarfs the singlet part in an extended part of parameter space adjoining the critical boundary that delimits the region where bound states exist, suggesting that $p$-wave character of the bound-state wave function should be prevalent there. The values $N_{SM}$ for individual triplet states are plotted in Fig. 3(b) for the parameter pairs $(\tilde{h}^{\mathrm{crit}}, \tilde{\epsilon}_0^{\mathrm{crit}})$ along the boundary of the region of existence for bound states in Fig. 1(a), given explicitly by Eq. (51). Asymptotically, as $\tilde{\epsilon}_0 \to 0$ and $\tilde{h} \to 1$, the bound-state wave function becomes the state $|1-1\rangle$. In contrast, the admixture of the $|1\,1\rangle$ triplet component to the bound-state wave function is vanishingly small, albeit not identically zero. Thus the dimer becomes an almost pure chiral triplet in the limit of weak interactions and close to the critical Zeeman energy $\tilde{h}^{\mathrm{crit}} \sim 1$.

## 3.2 Effect of finite center-of-mass momentum

We now look for solutions of the characteristic equation (48) for $\tilde{E}_{\mathrm{b}}$ with finite COM momentum $\mathbf{P}$. While we were not able to find closed-form analytic expressions for $F_{\mathbf{P}}(\tilde{E}_{\mathrm{b}}, \tilde{h})$ when $\mathbf{P} \neq \mathbf{0}$, numerical results for the bound-state energy and the fractional weights of the total-spin eigenstates from Eq. (37) are readily obtained. These turn out to not depend on the direction of $\mathbf{P}$, as the polar angle of $\mathbf{Q}$ either does not enter relevant mathematical expressions (e.g., the threshold $E_{\mathrm{th}}^{\mathrm{rel}}$ is a function of $\mathbf{Q}^2$) or can be absorbed into an integration variable (namely, the polar angle of $\mathbf{q}$) when it explicitly appears such as in Eq. (44) for $F_{\mathbf{P}}(\tilde{E}_{\mathrm{b}}, \tilde{h})$.

In Fig. 4(a), the binding energy for moderate interaction strength ($\tilde{\epsilon}_0 = 1$) is plotted as a function of both the Zeeman energy $h$ and the COM-momentum magnitude $|\mathbf{P}|$. We see that, as we discussed earlier in this paper, COM momentum acts qualitatively like an effective Zeeman coupling, with the region in the plane spanned by the variables $\tilde{h} \equiv h/(m\lambda^2)$ and $\tilde{P} \equiv |\mathbf{P}|/(m\lambda)$ where a bound state exists exhibiting an approximately circular symmetry. The boundary of this region is defined by the vanishing of the binding energy (23), i.e., $E_{\mathrm{b}} = E_{\mathrm{th}}^{\mathrm{rel}}$ with $E_{\mathrm{th}}^{\mathrm{rel}}$ from Eq. (22). The more stringent condition $E_{\mathrm{b}} < E_{\mathrm{th}}^{\mathrm{abs}}$ with $E_{\mathrm{th}}^{\mathrm{abs}}$ given in Eq. (21)

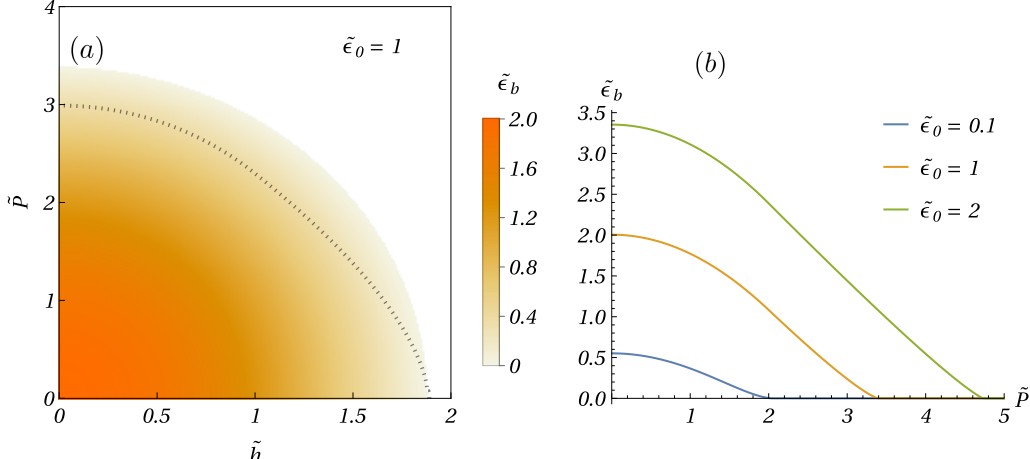

Figure 4: Effect of finite center-off-mass momentum **P** on dimer formation. Orange color in panel (a) indicates the range for dimensionless parameters quantifying the COM-momentum magnitude $[\tilde{P} \equiv |\mathbf{P}|/(m\lambda)]$ and the Zeeman energy $[\tilde{h} \equiv h/(m\lambda^2)]$ within which a 2D-fermion bound state exists. The color scale represents the dimensionless binding energy $\tilde{\epsilon}_b \equiv \epsilon_b/(m\lambda^2)$. The dotted black curve delimits the region of absolute bound-state stability where $E_b < E_{\text{th}}^{\text{abs}} \leq E_{\text{th}}^{\text{rel}}$, with the threshold energies defined in Eqs. (21) and (22). Data shown in panel (a) are obtained for a fixed value $\tilde{\epsilon}_0 \equiv \epsilon_0/(m\lambda^2) = 1$ of the dimensionless interaction strength. Panel (b) shows plots of $\tilde{\epsilon}_b$ as a function of $\tilde{P}$ for selected values of $\tilde{\epsilon}_0$ and fixed $\tilde{h} = 0$.

holds within the smaller region delimited by the dotted black curve, i.e., bound states are only metastable within the sliver of parameter space bounded by this curve and the boundary between orange and white regions in Fig. 4(a).

In Fig. 4(b), we plot the binding energy at zero Zeeman energy as a function of the dimensionless COM-momentum magnitude $\tilde{P}$ and observe a similar dependence as seen in Fig. 1(b) as a function of the Zeeman spin splitting. The weakening and eventual loss of the bound state with finite COM momentum is a well known property of dimers in the presence of spin-orbit coupling [29, 35]. It implies that, e.g., in a not fully condensed gas with a momentum distribution of a certain width, pairs at the outer edge of this distribution are no longer bound; thus such a setup would contain both bound pairs and unbound atoms. In a time-of-flight measurement, the unbound atoms are expected to form a ring around the bound pairs closer to the center of the momentum distribution [29].

As Fig. 4(a) shows, Zeeman splitting and COM momentum are qualitatively similar in their effect on the bound-state formation and its energy. However, details of the bound-state structure are quite different in the two cases. To illustrate this, we consider the fractional weights of total-spin eigenstates in the bound-state wave function, evaluating the integrals entering the expressions (37) numerically. In Fig. 5(a), we plot the difference $\Delta N$ between the combined weights of all triplet states and that of the singlet state as a function of $\tilde{h}$ and $\tilde{P}$ for fixed $\tilde{\epsilon}_0 = 0.1$. Triplet character is seen to be dominant all along the outer boundary of the parameter region where dimers are formed. However, the actual bound-state composition changes radically as one moves between the Zeeman-energy-dominated and the COM-momentum-dominated regimes. This becomes apparent when comparing Fig. 3(b) with Fig. 5(b), where we plot the individual fractional weights of total-spin eigenstates making up the bound state for $\tilde{h} = 0$. Here we see that, for finite COM momenta close to the boundary of the bound-state region, the triplet state $|10\rangle$ has the largest weight, whereas this state does not contribute at all to the zero-COM-momentum bound state (see Sec. 3.1). Thus the type of dominating

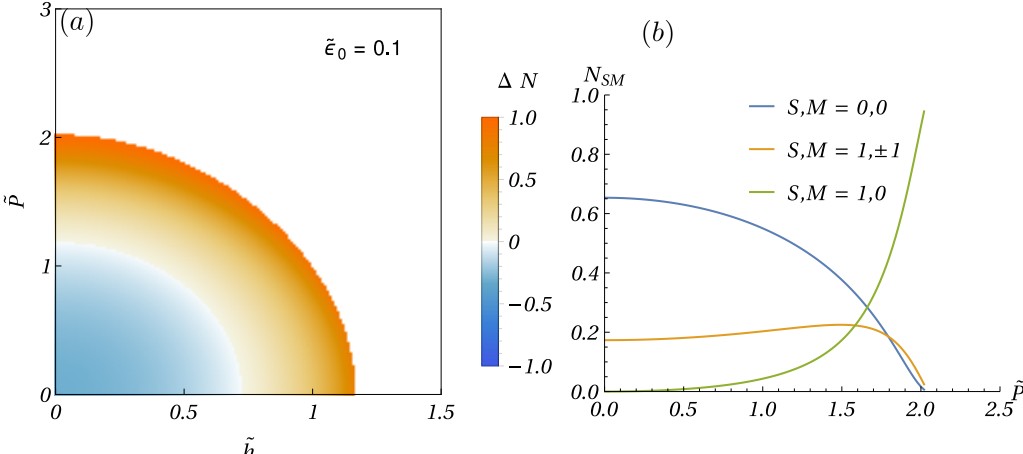

Figure 5: Bound-state triplet admixture for dimers with finite center-of-mass momentum $\mathbf{P}$. Panel (a) shows the quantity $\Delta N = \sum_M N_{1M} - N_{00}$, which measures the balance between triplet and singlet character in the 2D-fermion dimer, as a function of dimensionless COM-momentum magnitude $\tilde{P} \equiv |\mathbf{P}|/(m\lambda)$ and dimensionless Zeeman energy $\tilde{h} \equiv h/(m\lambda^2)$. The relative weights $N_{SM}$ of individual total-spin eigenstates contributing to the bound-state wave function are plotted in panel (b) as a function of $\tilde{P}$ for fixed $\tilde{h} = 0$, which is the parameter range along the vertical axis in panel (a). Data shown here were calculated for fixed dimensionless interaction strength $\tilde{\epsilon}_0 \equiv \epsilon_0/(m\lambda^2) = 0.1$.

triplet character differs crucially depending on how it is generated: large Zeeman splitting renders a spin-polarized triplet state to be dominant, whereas large COM-momentum favors the spin-unpolarized triplet state.

The behavior for large spin splitting of Zeeman or COM-motion origin can be contrasted with the case $\tilde{P} = 0$ and $\tilde{h} = 0$ that is also depicted in Fig. 5(b). In this situation, the singlet component to the bound state is dominant and the state $|10\rangle$ is completely absent. There is also a sizable triplet contribution, with the oppositely spin-polarized triplet states $|1 \pm 1\rangle$ contributing equally to preserve an overall spin-unpolarized bound-state wave function. The exact values for the total-spin-eigenstate proportions in the bound state for both COM momentum and Zeeman energy being zero, as well as their change as the Zeeman energy becomes finite, can be gleaned from analytical results provided in Appendix D.

## 4 Bound states formed with 1D-type spin-orbit coupling

The 1D-type spin-orbit coupling $\hat{\lambda}(\mathbf{p}) = \lambda p_x \hat{\sigma}_x$ is not isotropic since a particular in-plane direction is singled out. As a consequence, the two-particle relative-motion problem only depends on the $x$ component of the COM momentum $P_x$ through the effective magnetic field $\mathbf{B_P}$ of Eq. (13). Specifically, $P_x$ affects bound-state properties via the dependence of $F_{\mathbf{P}}(\tilde{E}_b, \tilde{h})$, $E_{th}^{rel}$ and the spinor amplitudes from Eqs. (35) on $\mathbf{Q}$, whose only nonzero component is $Q_x \equiv P_x$. The other COM component $P_y$ is only relevant for determining the metastability threshold $E_{th}^{abs}$ [see Eq. (21)].

Specializing the general formulae from Sec. 2 to the case with 1D-type spin-orbit coupling means adopting $\mathbf{B_P} = (\lambda P_x/2, 0, h)$, $\mathbf{Q} = (P_x, 0, 0)$ and $\mathbf{q} = (p_x, 0, 0)$. The resulting form of the implicit equation (48) for the bound-state energy can only be solved numerically. For the case of zero COM momentum (implying $P_x = 0$), we calculate the binding energy for the same

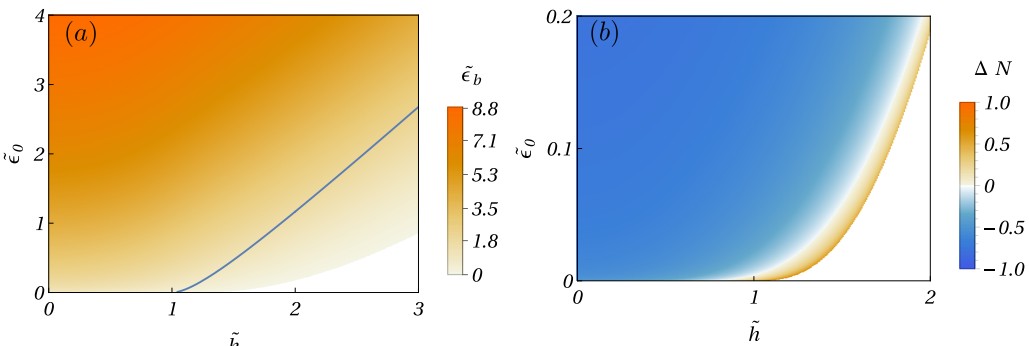

Figure 6: Dimensionless binding energy $\tilde{\epsilon}_b$ [panel (a)] and bound-state triplet character quantified by $\Delta N \equiv \sum_M N_{1M} - N_{00}$ [panel (b)] of 2D-fermion dimers with zero center-of-mass momentum formed in the presence of 1D-type spin-orbit coupling and Zeeman splitting. These plots can be compared with corresponding results for the 2D-type spin-orbit couplings shown in Figs. 1(a) and 3(a), respectively. The solid blue curve in panel (a) is the critical-boundary line (51) delimiting the region where a bound state exists for 2D-type spin-orbit coupling. Evidently, the existence region for bound states formed with asymmetric spin-orbit coupling extends well beyond.

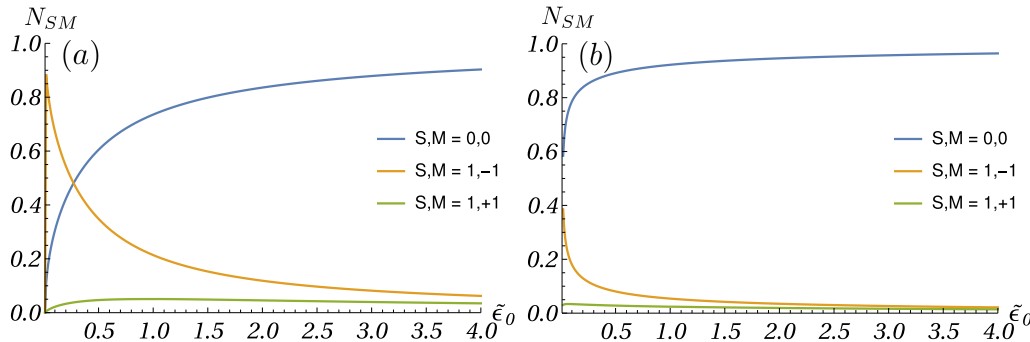

Figure 7: Fractional weights $N_{SM}$ of total-spin eigenstates in the 2D-fermion bound state with zero COM momentum ($\mathbf{P} = \mathbf{0}$) formed at fixed Zeeman energy $\tilde{h} = 1$, plotted as a function of the s-wave interaction strength parameterized by $\tilde{\epsilon}_0$ for 2D-type [panel (a)] and 1D-type [panel (b)] spin-orbit couplings. We do not show $N_{10}$ as it vanishes identically in both cases for zero COM momentum.

range of Zeeman-energy values and s-wave-interaction strengths as in the previous section. Results are shown in Fig. 6(a). We find that the parameter region within which a bound state exists is larger than in the case of 2D-type spin-orbit coupling. To illustrate this, the boundary line that we derived in Eq. (51) for the 2D-type case is drawn as the solid blue curve for comparison. In addition, the binding energy is generally higher than with 2D-type spin-orbit coupling.

We also adapt the formalism presented in Sec. 2.2 for calculating the fractional weights $N_{SM}$ of total-spin eigenstates in the bound state to the case of 1D-type spin-orbit coupling. This amounts to using $\mathbf{B}_\mathbf{P} = (\lambda P_x/2, 0, h)$, $\mathbf{Q} = (P_x, 0, 0)$ and $\mathbf{q} = (p_x, 0, 0)$ in Eqs. (35). In Fig. 6(b), the difference $\Delta N$ between the total weight from triplet states contributing to the bound state and the weight of the singlet state are shown. The results are qualitatively similar to the case with 2D-type spin-orbit coupling [compare Fig. 3(a)], but the region where the triplet contribution to the bound state dominates has a much narrower range in $\tilde{h}$. Figure 7 shows a comparison in the interaction-strength dependence of relative weights for the total-spin eigenstates present in bound states for 2D-type and 1D-type spin-orbit couplings. Again,

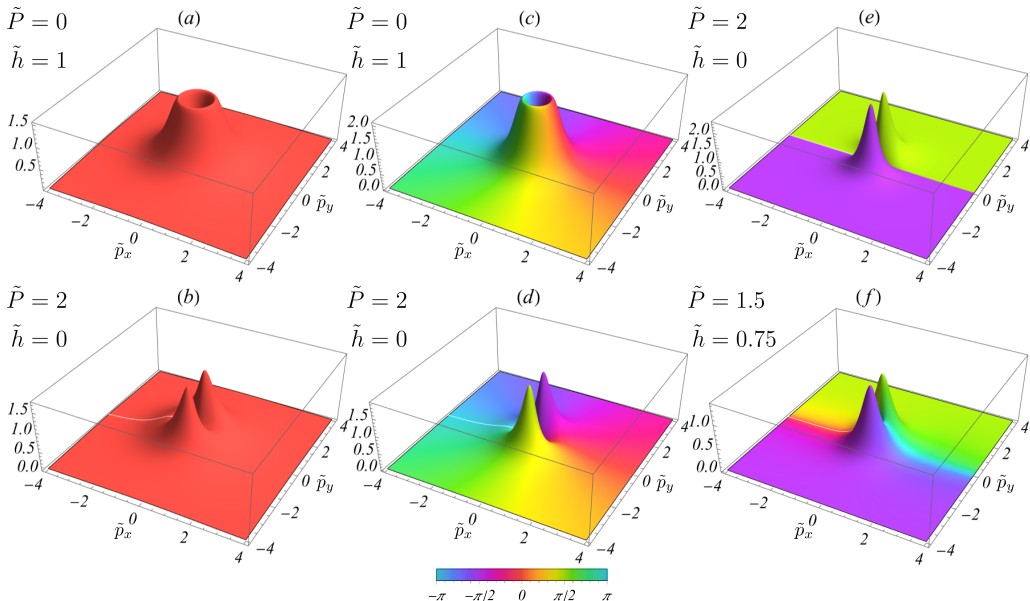

Figure 8: Orbital wave functions of the two-fermion bound state formed in the presence of 2D-Dirac spin-orbit coupling in relative-momentum $\mathbf{p} \equiv (p_x, p_y)$ representation. Surface height and color scale depict amplitude and phase, respectively, for $\langle S\,M|\psi_b(\mathbf{p})\rangle / N_\mathbf{P} \equiv \langle S\,M|\hat{G}_\mathbf{P}(E_b, \mathbf{p})|0\,0\rangle$ as a function of $\tilde{p}_j \equiv p_j/(m\,\lambda)$. Panels (a) and (b) show the singlet component ($S = 0$, $M = 0$) and panels (c) and (d) the spin-polarized triplet component ($S = 1$, $M = -1$), which is dominant for $\mathbf{P} = \mathbf{0}$, $\tilde{h} \approx 1$ and weak $s$-wave interaction strength. Panels (e) and (f) show the spin-unpolarized triplet component ($S = 1$, $M = 0$), which occurs only for finite $\mathbf{P}$. The dimensionless interaction strength is $\tilde{\epsilon}_0 = 0.2$, and values for the dimensionless Zeeman energy $\tilde{h}$ and COM momentum $\mathbf{P}/(m\,\lambda) = (\tilde{P}, 0)$ are indicated in each panel.

qualitatively similar behavior is exhibited in both cases, except that the region of dominant triplet character occurs at much weaker interaction strengths in the presence of 1D-type spin-orbit coupling. While 1D-type spin-orbit coupling is easier to realise experimentally than the 2D types [1], its utilization may pose new practical challenges due to the narrower parameter region where the triplet character dominates and the associated smallness of binding energies (typically a fraction of the spin-orbit energy scale $m\lambda^2$).

## 5  Orbital characteristics of the bound-state wave function

In previous sections, we have discussed the binding energy and the spin properties of two-particle bound states. We now explore features in the orbital part of the bound-state wave function. Specifically, we focus on the amplitudes $\langle S\,M|\psi_b(\mathbf{p})\rangle$ appearing in its expansion (28) in terms of total-spin eigenstates.

We plot representative examples obtained for 2D-Dirac spin-orbit coupling in Fig. 8. The singlet component, shown in panels (a) and (b), has no imaginary part. It exhibits radial symmetry in the relative-momentum ($\mathbf{p}$) plane for vanishing COM momentum [panel (a)]. A local minimum occurs at $\mathbf{p} = \mathbf{0}$ as a result of spin-orbit coupling modifying the noninteracting dispersion (16), creating an energy minimum at nonzero momentum. This local minimum in the orbital wave function associated with the singlet component at vanishing COM momentum disappears for sufficiently strong interactions $\tilde{\epsilon}_0 \geq 1$. For finite COM momentum $\mathbf{P}$ [the case

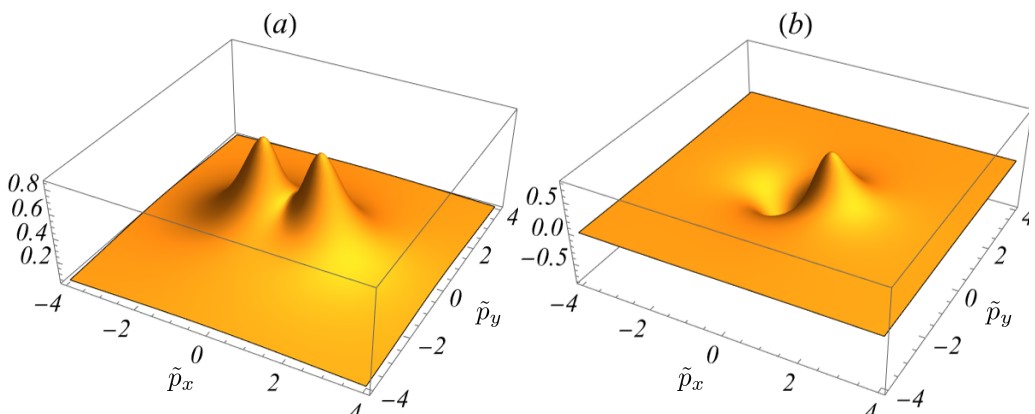

Figure 9: Orbital wave functions for two-fermion bound states formed in the presence of 1D-type spin-orbit coupling $\hat{\lambda}(\mathbf{p}) \equiv \lambda p_x \hat{\sigma}_x$ in the relative-momentum $\mathbf{p} \equiv (p_x, p_y)$ representation. Panel (a) shows the singlet $|0\,0\rangle$ component and panel (b) the $|1-1\rangle$ triplet component. The surface plots depict the real-valued functions $\langle S\,M|\psi_b(\mathbf{p})\rangle/N_\mathbf{P} \equiv \langle S\,M|\hat{G}_\mathbf{P}(E_b, \mathbf{p})|0\,0\rangle$, calculated with $\tilde{\epsilon}_0 = 0.01$ for the dimensionless interaction strength, Zeeman splitting $\tilde{h} = 1$ and vanishing COM momentum $\mathbf{P} = \mathbf{0}$, in their dependence on $\tilde{p}_j \equiv p_j/(m\lambda)$.

$\mathbf{P} \equiv (P, 0)$ is shown in panel (b)], the radial symmetry is broken as the singlet-wave-function amplitude gets suppressed along the direction of $\mathbf{P}$.

Panels (c) and (d) in Fig. 8 show the orbital part of the spin-polarized triplet state with $M = -1$ for the same set of parameters used in panels (a) and (b), respectively. This wave function is complex and, for $\mathbf{P} = \mathbf{0}$, shows typical $p$-wave behavior: a radially symmetric amplitude and single phase winding around the node at the origin $\mathbf{p} = \mathbf{0}$. For finite COM momentum, the phase still behaves the same, but the radial symmetry of the wave-function amplitude is broken in an analogous fashion as seen for the singlet component in panel (b). Panels (e) and (f) depict the orbital wave function for the spin-unpolarized triplet component ($S = 1$, $M = 0$) that is only ever finite for nonvanishing COM momentum. Without Zeeman splitting [panel (e)], this wave function is purely imaginary and proportional to the relative-momentum component perpendicular to the COM momentum [see the cross-product term in Eq. (35b)]. When the Zeeman energy is finite as well [panel (f)], the sign change in the imaginary part of the wave function turns into a full $2\pi$ phase rotation, while the node along the direction parallel to $\mathbf{P}$ softens into a finite local minimum.

It is straightforward to adapt the results plotted in Fig. 8 to the other 2D-type spin-orbit couplings. According to the general formulae given in Eqs. (35), the bound-state spinor amplitudes $\langle S\,M|\psi_b(\mathbf{p})\rangle$ are fundamentally a function of $\mathbf{q}$ defined in Eq. (4). As $\mathbf{q} \equiv (p_x, p_y, 0)$ for 2D-Dirac spin-orbit coupling (see the form of the matrix $\mathcal{M}$ given for this case in Table 1), the plots from Fig. 8 in fact directly show the momentum-space wave functions in their dependence on $q_x$ and $q_y$. Hence, the particular shape of the wave functions for any specific 2D-type spin-orbit coupling with its associated matrix $\mathcal{M}$ listed in Table 1 can be deduced by replacing the relative-momentum components in axes labels of plots from Fig. 8 according to the rule $p_a \to \sum_{\mu \in \{x,y\}} \mathcal{M}_{a\mu} p_\mu$. As the plots pertaining to finite $\mathbf{P}$ assumed the particular form $\mathbf{P} = (P, 0)$ and therefore represent $\mathbf{Q} = (P, 0, 0)$, they correspond to the case $\mathbf{P} = \mathcal{M}^T(P, 0, 0)^T$ for general 2D-type spin-orbit couplings.[2]

Results shown in Fig. 9 for 1D-type spin-orbit coupling look like anisotropic versions of

---

[2]This follows from multiplying both sides of the equation $\mathbf{Q} = \mathcal{M}\mathbf{P}$ from the left with $\mathcal{M}^T$ and applying the identity $\mathcal{M}^T\mathcal{M} = \mathbb{1}_{2\times2}$ that holds for all 2D-type spin-orbit couplings.

the behavior seen for 2D-Dirac spin-orbit coupling. As the amplitudes $\langle S M | \psi_b(\mathbf{p}) \rangle$ are all real-valued in this case, there is no phase winding but simply a sign inversion in the triplet component. Instead of a radially symmetric minimum, the singlet part exhibits a saddle point at $\mathbf{p} = \mathbf{0}$.

## 6 Experimental detection

Bound states could be probed with radio-frequency spectroscopy [20], or with high spectral resolution using magneto-association spectroscopy [42]. Spin-selective imaging can provide information on the spin content of ultra-cold atomic gases [43]. In order to determine the relative weight of different total-spin contributions to bound states, we propose to turn off the spin-orbit-coupling fields, which projects the cold-atom population to spin eigenstates, before using spin-selective imaging of single-particle populations. The *p*-wave character of the bound-state wave function can be detected by time-of-flight imaging of the single-particle momentum distribution. The characteristic signature of the *p*-wave character of the bound state is a vortex-like momentum distribution with a hole in the center, see Fig. 8. In the regime where triplet character dominates in the bound state, a density maximum is expected at the momentum scale $m \lambda \sqrt{\tilde{\epsilon}_b + \tilde{h}^2 - 1}$. As the square root is typically of order unity, this yields the characteristic momentum scale of spin-orbit coupling, which is well accessible in current experiments. The parameter regime with a large triplet component in the bound state could be probed by a fluorescence-imaging approach with single-atom spin and momentum resolution, as recently demonstrated [44]. Due to the single-particle resolution achieved in this experiment, it is possible to obtain relative momentum distributions at fixed COM momentum by post selection. The ratio of pairs with same and opposite spin would give a clear indication on the number of singlet or triplet pairs in the system.

## 7 Conclusions and outlook

In this paper, we have investigated the properties of bound states of two fermions in a 2D gas with Zeeman spin splitting and spin-orbit coupling. While spin-orbit coupling enhances binding, both the Zeeman splitting and a finite COM momentum of the dimer counteract the formation of bound states. We show that the COM momentum acts like an additional in-plane component of the Zeeman coupling. The bound state ceases to exist when either or both the Zeeman energy and the COM momentum exceed a threshold. For 1D-type spin-orbit coupling, the binding is stronger and the Zeeman energy for which a bound state can exist is larger than for 2D-type spin-orbit coupling.

Further, we have calculated the fractional weights of individual total-spin components in the bound state. In the systems we consider in this paper, there is a competition between the *s*-wave interactions, which project the two-body wave function onto the singlet state, and the spin-orbit coupling, which rotates the total-spin state into the triplet sector. By this mechanism, the triplet character of the wave function can become dominant. This happens when the Zeeman energy $|h|$ is near the critical value for the existence of a bound state. In this regime, the wave function is mostly in the spin-polarized triplet state that minimizes the total energy and has a *p*-wave-like shape with a node at zero relative momentum. We find that, for 1D-type spin-orbit coupling, this regime where triplet states dominate occurs in a much narrower range of Zeeman energies for fixed interaction strength (and *vice versa*) as compared to systems with 2D-type spin-orbit coupling. Nevertheless, large triplet-state fractions are still reached also for the bound states formed in the presence of 1D-type spin-orbit coupling.

With finite COM momenta, we find that the bound state also reaches dominant triplet character but now in the unpolarized $S = 1$, $M = 0$ triplet state. This triplet component is only present for nonzero COM momentum. These findings show that, in a many-body system such as a thermal Fermi gas, the distribution of COM momenta will lead to a gas with bound pairs in the singlet state at the center of the momentum distribution, triplet pairs further out, and unbound fermions at even higher momenta. We also discuss how such bound states could be detected experimentally; in particular, the detection of opposite-spin and same-spin correlations can reveal whether a 2D Fermi gas with spin-orbit coupling contains singlet or triplet bound pairs.

## Acknowledgements

We thank Chris Vale and Paul Dyke for helpful discussions.

**Funding information** This work was partially supported by the Marsden Fund of New Zealand (contract nos. VUW1713 and MAU2007) from government funding managed by the Royal Society Te Apārangi.

## A  Singlet and triplet projections of helicity-basis product states

This section provides useful identities involving the two-particle states (15) that are direct products of single-particle energy eigenstates labelled by the individual particles' momentum $\mathbf{p}_j$ and helicity $\alpha_j$. Indicating the spin-up (spin-down) eigenstate of $\hat{\sigma}_z$ by $|\!\uparrow\rangle$ ($|\!\downarrow\rangle$), we get for the singlet projection of such states

$$
\langle 0\,0|\alpha_1, \alpha_2\rangle_{\mathbf{p},\mathbf{P}} = \frac{1}{\sqrt{2}} \left( \langle\uparrow|\alpha_1,\mathbf{p}_1\rangle \langle\downarrow|\alpha_2,\mathbf{p}_2\rangle - \langle\downarrow|\alpha_1,\mathbf{p}_1\rangle \langle\uparrow|\alpha_2,\mathbf{p}_2\rangle \right)
$$

$$
= \frac{1}{\sqrt{2}} \left( \alpha_2 e^{-\frac{i}{2}(\phi_1-\phi_2)} \sqrt{\frac{Z_+ + \alpha_1 h}{2Z_+}} \sqrt{\frac{Z_- - \alpha_2 h}{2Z_-}} - \alpha_1 e^{\frac{i}{2}(\phi_1-\phi_2)} \sqrt{\frac{Z_+ - \alpha_1 h}{2Z_+}} \sqrt{\frac{Z_- + \alpha_2 h}{2Z_-}} \right).
$$
(52)

Analogously, for the overlap with the $S = 1$, $M = 0$ triplet state, we find

$$
\langle 1\,0|\alpha_1, \alpha_2\rangle_{\mathbf{p},\mathbf{P}} = \frac{1}{\sqrt{2}} \left( \langle\uparrow|\alpha_1,\mathbf{p}_1\rangle \langle\downarrow|\alpha_2,\mathbf{p}_2\rangle + \langle\downarrow|\alpha_1,\mathbf{p}_1\rangle \langle\uparrow|\alpha_2,\mathbf{p}_2\rangle \right)
$$

$$
= \frac{1}{\sqrt{2}} \left( \alpha_2 e^{-\frac{i}{2}(\phi_1-\phi_2)} \sqrt{\frac{Z_+ + \alpha_1 h}{2Z_+}} \sqrt{\frac{Z_- - \alpha_2 h}{2Z_-}} + \alpha_1 e^{\frac{i}{2}(\phi_1-\phi_2)} \sqrt{\frac{Z_+ - \alpha_1 h}{2Z_+}} \sqrt{\frac{Z_- + \alpha_2 h}{2Z_-}} \right).
$$
(53)

For the projections onto the spin-polarized triplet states, straightforward calculation yields

$$
\langle 1\,1|\alpha_1, \alpha_2\rangle_{\mathbf{p},\mathbf{P}} = \langle\uparrow|\alpha_1,\mathbf{p}_1\rangle \langle\uparrow|\alpha_2,\mathbf{p}_2\rangle = e^{-\frac{i}{2}(\phi_1+\phi_2)} \sqrt{\frac{Z_+ + \alpha_1 h}{2Z_+}} \sqrt{\frac{Z_- + \alpha_2 h}{2Z_-}},
$$
(54)

and

$$
\langle 1\,{-}1|\alpha_1, \alpha_2\rangle_{\mathbf{p},\mathbf{P}} = \langle\downarrow|\alpha_1,\mathbf{p}_1\rangle \langle\downarrow|\alpha_2,\mathbf{p}_2\rangle = e^{\frac{i}{2}(\phi_1+\phi_2)} \sqrt{\frac{Z_+ - \alpha_1 h}{2Z_+}} \sqrt{\frac{Z_- - \alpha_2 h}{2Z_-}}.
$$
(55)

The phases appearing in these identities are $\phi_j = \arg(p_{j,x} + i\, p_{j,y})$.

Relevant for calculations leading to results presented in this paper are the absolute square of the singlet projection and the latter's products with the triplet projections. To obtain more compact expressions, 3D vectors $\mathbf{Z}_\pm = (\lambda[Q_x/2 \pm q_x], \lambda[Q_y/2 \pm q_y], h)$ are introduced, in terms of which we find

$$|\langle 0\,0|\alpha_1, \alpha_2\rangle_{\mathbf{p},\mathbf{P}}|^2 = \frac{1}{4}\left(1 - \alpha_1\alpha_2 \frac{\mathbf{Z}_+ \cdot \mathbf{Z}_-}{Z_+ Z_-}\right), \tag{56}$$

$$\langle 1\,0|\alpha_1, \alpha_2\rangle_{\mathbf{p},\mathbf{P}\ \mathbf{p},\mathbf{P}}\langle \alpha_1, \alpha_2|0\,0\rangle = \frac{1}{4}\left(\frac{h(\alpha_1 Z_- - \alpha_2 Z_+)}{Z_+ Z_-} + i\,\alpha_1\alpha_2 \frac{(\mathbf{Z}_+ \times \mathbf{Z}_-)_z}{Z_+ Z_-}\right), \tag{57}$$

$$\langle 1\,1|\alpha_1, \alpha_2\rangle_{\mathbf{p},\mathbf{P}\ \mathbf{p},\mathbf{P}}\langle \alpha_1, \alpha_2|0\,0\rangle = \frac{1}{4\sqrt{2}}\left(\alpha_2 e^{-i\phi_2} \frac{\lambda|\mathbf{p}_2|\sqrt{(Z_+ + \alpha_1 h)^2}}{Z_+ Z_-}\right.$$
$$\left. - \alpha_1 e^{-i\phi_1} \frac{\lambda|\mathbf{p}_1|\sqrt{(Z_- + \alpha_2 h)^2}}{Z_+ Z_-}\right), \tag{58}$$

$$\langle 1\,{-}1|\alpha_1\,\alpha_2\rangle_{\mathbf{p},\mathbf{P}\ \mathbf{p},\mathbf{P}}\langle \alpha_1, \alpha_2|0\,0\rangle = \frac{1}{4\sqrt{2}}\left(\alpha_2 e^{i\phi_1} \frac{\lambda|\mathbf{p}_1|\sqrt{(Z_- - \alpha_1 h)^2}}{Z_+ Z_-}\right.$$
$$\left. - \alpha_1 e^{i\phi_2} \frac{\lambda|\mathbf{p}_2|\sqrt{(Z_+ - \alpha_2 h)^2}}{Z_+ Z_-}\right). \tag{59}$$

## B  Momentum representation of the Green's function

Here, we show how to obtain somewhat compact expressions for the momentum representation of the Green's function. As we consider only $s$-wave interactions that couple exclusively to the singlet channel, relevant formulae always contain the Green's function acting on the singlet total-spin eigenstate to the right. Thus we need to calculate the four matrix elements

$$\langle S\,M|\hat{G}_{\mathbf{P}}(E, \mathbf{p})|0\,0\rangle = \sum_{\alpha_1, \alpha_2} \frac{\langle S\,M|\alpha_1, \alpha_2\rangle_{\mathbf{p},\mathbf{P}\ \mathbf{p},\mathbf{P}}\langle \alpha_1, \alpha_2|0\,0\rangle}{E - \varepsilon_{\mathbf{P}}(\alpha_1, \alpha_2, \mathbf{p})}, \tag{60}$$

where we employed the Lehmann representation in terms of the eigenstates (14) of $\hat{H}_{\mathbf{P}}$.

Numerators appearing in (60) have been obtained in the previous section. As was first shown in Ref. [35], introducing the variable $s = \mathbf{p}^2/m - E$ allows us to perform the sum over the four combinations of $\{\alpha_1 = \pm, \alpha_2 = \pm\}$ and obtain a more compact form of the Green's function. For illustration, we show this in detail for the singlet component:

$$\langle 0\,0|\hat{G}_{\mathbf{P}}(E, \mathbf{p})|0\,0\rangle = \sum_{\alpha_1\alpha_2} \frac{|\langle 0\,0|\alpha_1, \alpha_2\rangle_{\mathbf{p},\mathbf{P}}|^2}{-(\alpha_1 Z_+ + \alpha_2 Z_-) - s} \tag{61}$$
$$= \frac{1}{4}\left[\left(1 - \frac{\mathbf{Z}_+ \cdot \mathbf{Z}_-}{Z_+ Z_-}\right)\left(\frac{1}{Z_+ + Z_- - s} + \frac{1}{-Z_+ - Z_- - s}\right)\right.$$
$$\left. + \left(1 + \frac{\mathbf{Z}_+ \cdot \mathbf{Z}_-}{Z_+ Z_-}\right)\left(\frac{1}{Z_+ - Z_- - s} + \frac{1}{-Z_+ + Z_- - s}\right)\right]$$
$$= -\frac{s}{2}\left[\frac{1}{s^2 - (Z_+ + Z_-)^2} + \frac{1}{s^2 - (Z_+ - Z_-)^2}\right.$$
$$\left. + \frac{\mathbf{Z}_+ \cdot \mathbf{Z}_-}{Z_+ Z_-}\left(\frac{1}{s^2 - (Z_+ - Z_-)^2} - \frac{1}{s^2 - (Z_+ + Z_-)^2}\right)\right]$$
$$= -\frac{s}{d}\left[s^2 - (Z_+^2 + Z_-^2 - 2\mathbf{Z}_+ \cdot \mathbf{Z}_-)\right]$$
$$= -\frac{s}{d}\left[s^2 - 4h^2 - \lambda^2(q_1^2 + q_2^2 + 2\mathbf{q}_1 \cdot \mathbf{q}_2)\right]$$

$$= -\frac{s}{d}\left[s^2 - 4h^2 - \lambda^2(\mathbf{q}_1 + \mathbf{q}_2)^2\right] = -\frac{s}{d}\left[s^2 - 4h^2 - \lambda^2\mathbf{Q}^2\right], \qquad (62)$$

thus obtaining (35a) with (36) giving the explicit expression for the denominator $d$. Along the same lines, the expressions (35b), (35c) and (35d) for Green's-function matrix elements involving the triplet states are derived. Our results agree with the expressions given in Ref. [35] for the vanishing-Zeeman-splitting limit $|h| \to 0$.

## C  Boundaries of parameter regions for P = 0 bound states

For 2D-type spin-orbit coupling and zero COM momentum, the boundary between the parameter regions with and without a bound state can be obtained analytically. In the following, we assume $|\tilde{h}| > 1$ since there is always a bound state when $|\tilde{h}| \leq 1$. To derive $\tilde{\epsilon}_0^{\text{crit}}$, we substitute the threshold energy $\tilde{E}_{\text{th}} = -2|\tilde{h}|$ applicable for $|\tilde{h}| > 1$ in place of $\tilde{E}_{\text{b}}$ into Eq. (48), using also the analytical expression (49) for $F_0(\tilde{E}_{\text{b}}, \tilde{h})$. First we consider

$$\begin{aligned}
F_0(\tilde{E}_{\text{b}}, \tilde{h})\big|_{\tilde{E}_{\text{b}} = -2|\tilde{h}|} = \lim_{\tilde{E}_{\text{b}} \to -2|\tilde{h}|} &\left\{ \frac{-\tilde{E}_{\text{b}} \ln(-\tilde{E}_{\text{b}})}{2(\tilde{h}^2 + \tilde{E}_{\text{b}})} \right. \\
&- \frac{\left(1 + \sqrt{1 + \tilde{h}^2 + \tilde{E}_{\text{b}}}\right)\left(-\tilde{E}_{\text{b}} - 2 + 2\sqrt{1 + \tilde{h}^2 + \tilde{E}_{\text{b}}}\right)}{4\sqrt{1 + \tilde{h}^2 + \tilde{E}_{\text{b}}}\,(\tilde{h}^2 + \tilde{E}_{\text{b}})} \ln\left(-\tilde{E}_{\text{b}} - 2 + 2\sqrt{1 + \tilde{h}^2 + \tilde{E}_{\text{b}}}\right) \\
&\left. + \frac{\left(1 - \sqrt{1 + \tilde{h}^2 + \tilde{E}_{\text{b}}}\right)\left(-\tilde{E}_{\text{b}} - 2 - 2\sqrt{1 + \tilde{h}^2 + \tilde{E}_{\text{b}}}\right)}{4\sqrt{1 + \tilde{h}^2 + \tilde{E}_{\text{b}}}\,(\tilde{h}^2 + \tilde{E}_{\text{b}})} \ln\left(-\tilde{E}_{\text{b}} - 2 - 2\sqrt{1 + \tilde{h}^2 + \tilde{E}_{\text{b}}}\right) \right\}.
\end{aligned} \qquad (63)$$

Recognizing that the third term between curly brackets on the r.h.s. of Eq. (63) vanishes in the limit $\tilde{E}_{\text{b}} \to -2|\tilde{h}|$ with $|\tilde{h}| > 1$, we obtain

$$F_0(-2\tilde{h}, \tilde{h}) = \frac{\ln(2|\tilde{h}|) - \ln[4(|\tilde{h}| - 1)]}{|\tilde{h}| - 2} \equiv \frac{-1}{|\tilde{h}| - 2} \ln\left(2\frac{|\tilde{h}| - 1}{|\tilde{h}|}\right). \qquad (64)$$

Using this, Eq. (48) with $\tilde{E}_{\text{b}} \to -2|\tilde{h}|$ becomes

$$\gamma + \ln\left(\frac{1}{2}\sqrt{\frac{2|\tilde{h}|}{\tilde{\epsilon}_0}}\right) \equiv \frac{1}{2}\ln\left(\frac{e^{2\gamma}|\tilde{h}|}{2\tilde{\epsilon}_0}\right) = \ln\left(2\frac{|\tilde{h}| - 1}{|\tilde{h}|}\right)^{\frac{1}{|\tilde{h}| - 2}}, \qquad (65)$$

which can be straightforwardly solved for $\tilde{\epsilon}_0$ to yield Eq. (51).

In a similar fashion, we determine the boundary line dividing regions in the $\tilde{\epsilon}_0$-$\tilde{h}$ parameter space where $\tilde{E}_{\text{b}} \leq -1 - \tilde{h}^2$ and $-1 - \tilde{h}^2 < \tilde{E}_{\text{b}} < -2|\tilde{h}|$, i.e., the curve where $\tilde{E}_{\text{b}} = -1 - \tilde{h}^2$ for $|\tilde{h}| > 1$. Introducing $\tilde{\delta}_{\text{b}} \equiv -1 - \tilde{h}^2 - \tilde{E}_{\text{b}}$, we find

$$F_0(-1 - \tilde{h}^2, \tilde{h}) =$$
$$\frac{1}{4}\left\{ 2\left(\tilde{h}^2 - 1\right) \underbrace{\lim_{\tilde{\delta}_{\text{b}} \to 0} \frac{1}{\sqrt{\tilde{\delta}_{\text{b}}}}\left[\frac{\pi}{2} + \arctan\left(\frac{1 - \tilde{h}^2}{2\sqrt{\tilde{\delta}_{\text{b}}}}\right)\right]}_{\frac{2}{\tilde{h}^2 - 1}} - \left(1 + \tilde{h}^2\right)\left[2\ln\left(1 + \tilde{h}^2\right) - \ln\left(1 - \tilde{h}^2\right)^2\right] \right\}$$
$$= 1 - \frac{1 + \tilde{h}^2}{2}\left[\ln\left(1 + \tilde{h}^2\right) - \ln\left|\tilde{h}^2 - 1\right|\right]. \qquad (66)$$

With this (using also $|\tilde{h}^2-1| \equiv \tilde{h}^2-1$ with our assumptions), Eq. (48) for $\tilde{E}_b = -1-\tilde{h}^2$ becomes

$$\gamma + \ln\left(\frac{1}{2}\sqrt{\frac{1+\tilde{h}^2}{\tilde{\epsilon}_0}}\right) = -1 - \frac{1+\tilde{h}^2}{2}\,\ln\left(\frac{\tilde{h}^2-1}{\tilde{h}^2+1}\right), \tag{67}$$

yielding Eq. (50).

## D  Analytical results for fractional weights of total-spin eigenstates in the bound state for $P = 0$ and $\tilde{E}_b \leq -1-\tilde{h}^2$

To calculate the fractional weights $N_{SM}$ of total-spin eigenstates in the two-particle bound states according to Eq. (37), integrals over the squared magnitude of Green's-function matrix elements are needed. For the case of vanishing COM momentum and bound-state energy satisfying $\tilde{E}_b \leq -1-\tilde{h}^2$, we can provide analytical results for the latter:

$$\int d^2p \, |\langle 0\,0|\hat{G}_0(E_b, \mathbf{p})|0\,0\rangle|^2 = \frac{\tilde{h}^2(\tilde{E}_b - 2) + \tilde{E}_b^2}{2(\tilde{h}^2 + \tilde{E}_b)^2(-1-\tilde{h}^2-\tilde{E}_b)}$$
$$- \frac{\tilde{h}^4\left(\ln(\tilde{E}_b^2 - 4\tilde{h}^2) - 2\ln(-\tilde{E}_b)\right)}{(\tilde{h}^2 + \tilde{E}_b)^3} - \frac{\tilde{h}^4}{\tilde{E}_b(\tilde{h}^2 + \tilde{E}_b)^2}$$
$$- \frac{\left(6\tilde{h}^2\tilde{E}_b^3 + 4\tilde{h}^6(\tilde{E}_b - 3) + \tilde{h}^4(3\tilde{E}_b(3\tilde{E}_b - 4) - 8) + \tilde{E}_b^4\right)\left(\arctan\left(\frac{\tilde{E}_b + 2}{2\sqrt{-1-\tilde{h}^2-\tilde{E}_b}}\right) + \frac{\pi}{2}\right)}{4(-1-\tilde{h}^2-\tilde{E}_b)^{3/2}(\tilde{h}^2 + \tilde{E}_b)^3}, \tag{68}$$

and

$$\int d^2p \, |\langle 1\,\pm1|\hat{G}_0(E_b, \mathbf{p})|0\,0\rangle|^2 = \frac{2((\tilde{h}\mp1)\tilde{h}+1)\tilde{h}^2 + (3\tilde{h}\mp2)\tilde{h}\tilde{E}_b + \tilde{E}_b^2}{4(\tilde{h}^2 + \tilde{E}_b)^2(-1-\tilde{h}^2-\tilde{E}_b)} - \frac{\tilde{h}^2}{2(\tilde{h}^2 + \tilde{E}_b)^2}$$
$$+ \Big( \pm4\tilde{h}^7 + \tilde{h}^4(9\tilde{E}_b^2 + 6\tilde{E}_b - 4) + 4\tilde{h}^6(\tilde{E}_b - 1) \pm 6\tilde{h}^5\tilde{E}_b$$
$$\mp 4\tilde{h}^3\tilde{E}_b + 2\tilde{h}^2\tilde{E}_b(3\tilde{E}_b(\tilde{E}_b + 2) + 2) \mp 2\tilde{h}\tilde{E}_b^2(\tilde{E}_b + 2)$$
$$+ \tilde{E}_b^3(\tilde{E}_b + 2)\Big) \frac{\arctan\left(\frac{\tilde{E}_b + 2}{2\sqrt{-1-\tilde{h}^2-\tilde{E}_b}}\right) + \frac{\pi}{2}}{8(-1-\tilde{h}^2-\tilde{E}_b)^{3/2}(\tilde{h}^2 + \tilde{E}_b)^3}$$
$$- \frac{\tilde{h}\left(\tilde{h}^3 - (1\mp\tilde{h})\tilde{h}\tilde{E}_b \pm \tilde{E}_b^2\right)\left(2\ln(-\tilde{E}_b) - \ln(\tilde{E}_b^2 - 4\tilde{h}^2)\right)}{4(\tilde{h}^2 + \tilde{E}_b)^3}. \tag{69}$$

Note that, because $\langle 1\,0|\hat{G}_0(E_b, \mathbf{p})|0\,0\rangle = 0$, $N_{10} = 0$ for the case $\mathbf{P} = 0$.

For the case of vanishing Zeeman splitting, the results above simplify considerably, leading to

$$\int d^2p \, |\langle 0\,0|\hat{G}_0(E_b, \mathbf{p})|0\,0\rangle|^2 \to \frac{1}{2(-1-\tilde{E}_b)} - \frac{\tilde{E}_b\left(\arctan\left(\frac{\tilde{E}_b + 2}{2\sqrt{-1-\tilde{E}_b}}\right) + \frac{\pi}{2}\right)}{4(-1-\tilde{E}_b)^{3/2}}, \tag{70}$$

$$\int d^2p \, |\langle 1\,\pm1|\hat{G}_0(E_b, \mathbf{p})|0\,0\rangle|^2 \to \frac{1}{4(-1-\tilde{E}_b)} + \frac{(\tilde{E}_b + 2)\left(\arctan\left(\frac{\tilde{E}_b + 2}{2\sqrt{-1-\tilde{E}_b}}\right) + \frac{\pi}{2}\right)}{8(-1-\tilde{E}_b)^{3/2}}. \tag{71}$$

Using $\tilde{E}_b = -1.55$ [consistent with $\tilde{\epsilon}_b = 0.55$ obtained for $\tilde{\epsilon}_0 = 0.1$, $\tilde{h} = 0$ and $\tilde{P} = 0$; see Fig. 4(b)] in the expressions (70) and (71), one derives $N_{00} = 0.65$ and $N_{1\pm1} = 0.17$ in agreement with the $\tilde{P} = 0$ results shown in Fig. 5(b).

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
