# Peer review of "Triplet character of 2D-fermion dimers arising from $s$-wave attraction via spin-orbit coupling and Zeeman splitting"

_SciPost Physics, doi:SciPost Phys. 12, 167 (2022)_

## Round 1 · Referee Report · Peter Schmelcher (Referee 1) · 2022-2-7

**Referee Report on: 2112.09336v1**

In the manuscript titled "Triplet character of 2D-fermion dimers arising from s-wave attraction via spin-orbit coupling and Zeeman splitting" the authors investigate a fermionic two-component mixture that interacts via short-range interactions, where the particles' orbital motion is coupled to their spin degree of freedom via spin-orbit coupling. The central finding of this investigation is a two-particle bound state with a triplet character. This result is determined from the observation of the eigenstates from the corresponding Schrödinger equation in combination with Bethe-Peierls boundary conditions. The manuscript is mostly well-written and the results look promising. Therefore, I think that it satisfies the requirement for a publication in SciPost Physics. However, I suggest minor adaptions to increase the comprehensibility of the presented results. In the following part, I elaborate on the main points that should be addressed in the revised manuscript and provide some additional comments for possible improvements.

**Main points**

a. In section 2 the different kinds of spin-orbit coupling are presented in terms of a transformation matrix $\mathcal{M}$. I think that already at this stage it should be made clear to the reader that the $\mathcal{M}$ matrix possesses the property $\mathcal{M}\mathcal{M}^T = \mathbb{1}$ and hence $\boldsymbol{q}^2 = \boldsymbol{p}^2$ in the case of 2D spin-orbit coupling. In contrast, in the 1D case, the above does not hold and one has $\boldsymbol{q}^2 = p_x^2$. This property elucidates the differences between 1D and 2D spin-orbit coupling and has important ramifications later on, as for instance, it can explain why the different forms of 2D spin-orbit coupling result in the same bound state energy as noted in section 3. Based on the above, I have a few suggestions on how the authors can rectify this issue, which I outline below.

   1. A brief discussion in section 2 should be added about the above-mentioned properties of the $\mathcal{M}$ matrix.

   2. An explanation, in the beginning of section 3, which addresses that $\mathcal{M}\mathcal{M}^T = \mathbb{1}$ leads to the bound-state energies being independent of the precise form of the 2D spin-orbit coupling would be useful. In addition, it should be explained that the same is not true regarding the wavefunctions which are sensitive to the form of $\mathcal{M}$.

   3. On a related note, in section 5 the authors write "It is possible to transform these results into those applicable to other 2D-type spin-orbit couplings by swapping and/or mirroring the relative-momentum axes according to the structure of $\mathcal{M}$-matrices given in Table. 1". However, no explicit formula is given. Therefore given these issues, the authors should provide an additional appendix where they demonstrate how the wavefunction transforms for the different 2D spin-orbit coupling types.

b. In Eq. (5) the term $\mathbf{B_P} \cdot \boldsymbol{\Sigma}$ is introduced which is then briefly discussed. Considering the results at the end of section 3.2, I think a more to-the-point discussion regarding the properties of this term would be helpful. As a guide to the authors, in the following, I provide a possible interpretation of the results of Fig. 4(b), based on Eq. (5), which they can verify or falsify based on their data and then update the discussion in sections 2 and 3.2 accordingly. First, notice that the above-mentioned term can be rewritten as follows

$$\mathbf{B_P} \cdot \boldsymbol{\Sigma} = \frac{\lambda \mathbf{Q}^2}{2}(\cos(\theta)\hat{\Sigma}_x + \sin(\theta)\hat{\Sigma}_y) + h\hat{\Sigma}_z \qquad (1)$$

where $\theta$ depends only on the direction of the COM momentum and the specific type of spin-orbit coupling. Therefore, it is obvious that $\lambda \mathbf{Q}^2$ acts as an effective Rabi-coupling among the spin states and $h$ is the corresponding Zeeman energy. As a consequence, large amplitudes of the COM momentum favour polarization along the axis $\mathbf{n} = (\cos\theta, \sin\theta, 0)$ lying on the $xy$ plane. Large $h$ give rise to the polarization of the system along the $z$ spin-axis. The above explains the high $\mathbf{P}$ results of Fig. 4(b) since for a two-particle system they imply that for large $\mathbf{Q}^2 = \mathbf{P}^2$ the system is led towards the superposition states $|1\mathbf{n}_\pm\rangle \equiv e^{I(-\sin\theta\hat{\Sigma}_x + \cos\theta\hat{\Sigma}_y)\pi}|1 \pm 1\rangle$ and therefore a population of $N_{10} \sim 0.5$ and $N_{1\pm1} \sim 0.25$ is expected. The deviation of the populations of $N_{1\pm1}$ from the above value can be explained by the presence of the term $\mathbf{q} \cdot (\boldsymbol{\sigma} \otimes \mathbb{1} - \mathbb{1} \otimes \boldsymbol{\sigma})$, which breaks the SU(2) symmetry associated with $S$ and couples the states $|1 \pm 1\rangle$ and $|00\rangle$.

c. The derivations in section 2.2 are confusing in my opinion. This particularly applies to those relating to Eq. (23). More specifically, one can easily verify that $c_{\mathbf{P}}V_0$ is the normalization

factor of $|\psi_b\rangle\rangle$ by substituting Eq. (17) and Eq. (21) into $\langle\langle\psi_b|\psi_b\rangle\rangle$ and noticing that in the case of pure $s$-wave interactions the integral over momenta of $\langle SM|\psi_p(\mathbf{p})\rangle$ vanishes for all cases with $S \neq 0$. It is important that the authors comment on this. As it stands, Eq. (23) cannot be used to evaluate $c_\mathbf{P}$ as by substituting Eq. (22) to Eq. (23), the latter reduces to the characteristic equation, Eq. (28). By mentioning that $c_\mathbf{P} V_0$ is the normalization factor of $|\psi_b\rangle\rangle$ it makes clear why it is independent of $\mathbf{P}$ and why it does not contribute to Eq. (27) enhancing the readability of this section. The same change also makes clear to the reader that $|\psi_b(\mathbf{p})\rangle$ are not orthonormal, a fact which is not mentioned in the current version of the manuscript. In addition, it lifts the ambiguity on how the orbital wavefunctions are calculated in section 5.

d. For the derivation of Eq. (31) the completeness of the two-body helicity spin-basis,

$$\sum_{\alpha_1,\alpha_2} |\alpha_1,\alpha_2\rangle_{\mathbf{p},\mathbf{P}}\,{}_{\mathbf{p},\mathbf{P}}\langle\alpha_1,\alpha_2| = \hat{\mathbb{1}}_{\mathbf{p},\mathbf{P}}, \tag{2}$$

has to be employed. A property that is not mentioned in the current form of the manuscript. Here $\hat{\mathbb{1}}_{\mathbf{p},\mathbf{P}}$ refers to the unity operator within the subspace of fixed relative, $\mathbf{p}$, and center-of-mass, $\mathbf{P}$ momenta. To make the derivation more transparent this property should be added in section 2.1 where the two-body helicity basis is first introduced. This comment should be subsequently referred to in the discussion regarding Eq. (31).

**Minor comments**

1. In Eq. (2) $h$ is introduced but is not defined as denoting the Zeeman energy. The authors should make sure that this quantity is defined in the updated version of their manuscript. In addition, they might consider changing the notation from $h$ to $\Delta$ to avoid any possible confusion with the Planck constant.

2. In the discussion following Eq. (3), it might be useful to note that $\mathcal{M}$ provides a transformation from the configuration to spin space. Moreover, the authors might consider amending the expression regarding $\hat{\lambda}(\mathbf{p})$ by including the expansion of the matrix multiplications, $\hat{\lambda}(\mathbf{p}) = \lambda \sum_{a=\{x,y,z\}} \sum_{\mu\in\{x,y\}} \sigma_a M_{a\mu} p_\mu$. This change will make the distinction between spatial coordinates $\mu$ and spin coordinates $a$ clear.

3. Regarding Eq. (10) it should be commented that $\alpha_1$ and $\alpha_2$ take values from -1 to 1 and describe the helicity states.

4. In Eq. (23) it might be preferable to denote $c_\mathbf{P}$ as a function $c(\mathbf{P})$, or even as $N(\mathbf{P})$, where the latter points out its role as the normalization constant.

5. The term "secular equation", might be unfamiliar to some physicists, it might be a good idea to replace it with some more familiar term characteristic or eigenvalue equation.

6. In Eq. (29) the transformation from momentum to real space is only done for the relative coordinate while keeping the COM momentum constant, I think that also the latter fact should be mentioned.

7. During the discussion regarding the reduction to dimensionless quantities, it should be noted that this procedure results in the various quantities mentioned thereafter being measured in the characteristic units of the spin-orbit coupling strength, i.e. $\hbar = m = \lambda = 1$.

8. When the dimensionless threshold energy $\tilde{E}_\mathrm{th}$ is introduced in section 3.1, Eq. (15), which defines this quantity, should also be mentioned as a reminder.

9. Additionally, at the beginning of section 3.2 the "quantity appearing on the r.h.s. of Eq. (37)" should be replaced by the concrete specification of the quantity, namely, $F_\mathbf{P}(\epsilon_0, h)$ for $\mathbf{P} \neq 0$.

10. The quantity $\tilde{P}$ is undefined within the main text, with its definition given only in the caption of Fig. 3. The authors should also define it where it first appears in the main text.

11. During the discussion in section 4, it remains unresolved how the $P_y$ component of the COM affects the results and one has to trace back to Eq. (8) and (9) to verify that it contributes to a $P_y^2/(2m)$ shift of the bound state energy. The authors should briefly comment on this in section 4.

12. In the end of section 4, to emphasize the fact that "This is important for experiments, as 1D-type spin-orbit coupling is easier to realize, but tuning the interactions to a sufficiently weak strength (e.g. near the zero-crossing of a Feshbach resonance) might prove challenging.", it would be important to include the appropriate citations.

13. A brief discussion about the envisaged experimental probes should be included in the introduction of the manuscript.

14. In section 6, which describes the possible experimental detection, a criterion referring to a characteristic momentum scale $m\lambda\sqrt{\tilde{\epsilon}_b + \tilde{h}^2 - 1}$ is derived. The authors should briefly comment on how this momentum scale compares to the resolution that is experimentally achievable considering the typical temperatures in state-of-the-art ultracold atom experiments.

---

## Round 1 · Referee Report · Anonymous (Referee 2) · 2022-3-22

Strengths

  1. This study is well-motivated and timely.

  2. Interesting new results on two-body bound states of spin-orbit coupled fermions.

  3. The results will be useful as a starting point for developing many-body theories of SO-interacting fermions.

Weaknesses

Presentation lack sufficient detail (see requested changes below).

Report

This manuscript presents a theoretical study of interacting spin-1/2 fermions in two dimensions. The interactions are composed of isotropic short-range attraction and the spin-orbit coupling, and the Zeeman interactions. The authors find triplet bound states which could be observed in the regime of sufficiently strong spin-orbit (SO) coupling, which leads to interesting unconventional pairing mechanisms. These mechanisms could give rise to topological superfluidity with Majorana-fermion excitations. This study is therefore well-motivated and timely.

The major new aspect of this work is the inclusion of the Zeeman splitting on the same footing as center-of-mass momentum, which has apparently not been done before. In addition, the authors examine the spin properties of the bound state in detail and consider a usefully wide range of SO couplings, such as Dirac, Rashba, and Dresselhaus in 2D and the p_x \sigma_x coupling in 1D. They first solve the two-body problem with a general SO coupling to obtain the wavefunction |\psi_b(p)> and then project the solution onto the total spin states of the two-particle system (Eqs. 25). They finally consider the different types of SO couplings and analyze the properties of the bound states obtained with each type of coupling.

Some of the interesting results obtained by the authors include: (i) the COM momentum acts as an effective magnetic field and (ii) the two-body bound states disappear when the COM momentum exceeds a certain threshold.

Perhaps even more importantly, the results obtained in this manuscript will be useful as a starting point for developing many-body theories of SO-interacting fermions. Thus, in my opinion, the Expectation “Open a new pathway in an existing or a new research direction, with clear potential for multipronged follow-up work” is satisfied.

It should also be noted that the manuscript is clearly and concisely written. The general acceptance criteria will also be satisfied after the authors address my comments below.
As such, I recommend this manuscript for publication in SciPost Physics.

Requested changes

  1. The parameter b defined below Eq. (5) gives the relative strength of the Zeeman and SO interaction seems to have the dimension of momentum. Does this parameter then adequately reflects the relative strength of these Zeeman and SO interactions. Would it not be better to use a dimensionless ratio of interaction strengths?

  2. Equations (9) establish that the relative motion of two particles in the COM frame depends on the COM momentum P. This is an unusual situation because the reason one introduces the COM and relative coordinates in the first place is to decouple the Hamiltonian into two commuting parts (the COM and internal Hamiltonians).

Hence, two points should be clarified. First, is there really any practical advantage to using the coordinates in Eq. (6)? Second, it would be helpful to mention exactly which interactions are responsible for the coupling between the external and internal degrees of freedom. Are these the SO coupling terms of the kind \sigma_x p_y?

  1. In Section 2.2 the authors use the Green function approach to obtain the bound states of s-wave interacting particles. Is this the only approach that can be used? It would also be helpful for the general reader to understand how this approach works using a simple example. A reference to the approach being applied to, e.g., two interacting particles in the absence of the SO interaction, would be helpful.

---

## Round 2 · Author Response

We thank both referees for their detailed comments and constructive suggestions that have helped to improve the readability of our manuscript. Following the referees’ comments, and also noticing inconsistencies in the treatment of the bound state threshold in the original manuscript, we have revised the manuscript substantially. The resulting changes to the numerical results and also to formulas, e.g. for the bound state thresholds, now show an enlarged parameter regime for the existence of bound states as well as enlarged parameter regimes where the triplet character of bound states could be observed. Thus the new results confirm and, at times, strengthen the conclusions of the original manuscript.
We elaborate further below on how we have incorporated the referees’ suggestions into the revised version. For convenience, we repeat passages from each individual report before commenting on them. At the very end of this reply, we discuss additional revisions made in the manuscript that were not prompted by the referee reports.
Response to Report 1 and associated changes:
We are grateful to Prof Schmelcher for the positive evaluation of our work, his stimulating thoughts, and the many concrete suggestions for improving the readability of our manuscript. The itemized Main points and Minor comments from the report have been addressed as follows:
Main points:
a. “I think that already at this stage it should be made clear to the reader that the matrix M possesses the property M M^T = 1 and hence q^2 = p^2 in the case of 2D spin-orbit coupling…”
We thank the referee for this helpful suggestion, which we have picked up as detailed below.
a.1.: "A brief discussion in section 2 should be added about the above-mentioned properties of the M matrix."
Below Eq. (4), we have added the sentence ‘The fact that ... diag(1,0).’ to discuss these properties of the M matrix and consequences for 2D-type vs. 1D-type spin-orbit couplings.
a.2.: "An explanation, in the beginning of section 3, which addresses that M M^T = 1 leads to the bound-state energies being independent of the precise form of the 2D spin-orbit coupling would be useful. In addition, it should be explained that the same is not true regarding the wavefunctions which are sensitive to the form of M."
The preamble of section 3 has been extended to discuss these important points in detail. Specifically, we added the part ‘because they give ... [see Eqs. (35)].’
a.3.: "On a related note, in section 5 the authors write ‘It is possible to transform these results into those applicable to other 2D-type spin-orbit couplings by swapping and/or mirroring the relative-momentum axes according to the structure of M-matrices given in Table. 1’. However, no explicit formula is given. Therefore given these issues, the authors should provide an additional appendix where they demonstrate how the wavefunction transforms for the different 2D spin-orbit coupling types."
We found that it wasn’t necessary to relegate these details to an additional appendix, as they can be written down quite compactly. Hence, we have added the detailed description asked for by the referee directly in section 5 as the new penultimate paragraph ‘It is straightforward ... spin-orbit couplings.^2’, including the explanatory footnote 2.
b. "In Eq. (5) [sic: (9)?] the term B_P.Sigma is introduced which is then briefly discussed. Considering the results at the end of section 3.2, I think a more to-the-point discussion regarding the properties of this term would be helpful. As a guide to the authors, in the following, I provide a possible interpretation of the results of Fig. 4(b), based on Eq. (5) [sic: (9)?], which they can verify or falsify based on their data and then update the discussion in sections 2 and 3.2 accordingly."
We thank the referee for sharing his ideas about how the form of the spin-splitting vector B_P could be interpreted, and how this interpretation may explain numerical results for the triplet-state admixture in dimer states. After thinking this through, however, we ultimately decided that treating the Q-dependent in-plane part of B_P as a Rabi coupling is not useful in the general case, especially if the Zeeman energy h is small or absent. Also, the interplay of the Zeeman term B_P.Sigma and the term proportional to lambda q [see new Eq. (12), which was Eq. (9) in the previous version] appears to be less trivial than envisioned by the referee. This is particularly apparent from the new results for the triplet admixture in dimers [new Fig. 5(b)] obtained after correcting an issue with the threshold energy of two-particle bound states (see detailed discussion below). While these considerations have prevented us from adopting the referee’s proposed viewpoint, we decided to follow the suggestion to discuss effects arising from B_P in greater detail right after this quantity is introduced [new Eq. (13)]; specifically in the passage "The last term … or h vanish.". As part of this discussion, we foreshadow similarities and differences between how finite h and finite P affect bound-state properties.
c.: "The derivations in section 2.2 are confusing in my opinion. This particularly applies to those relating to Eq. (23). More specifically, one can easily verify that c_P V_0 is the normalization factor of |psi_b>> ... It is important that the authors comment on this. … The same change also makes clear to the reader that |psi_b(p)> are not orthonormal, a fact which is not mentioned in the current version of the manuscript. In addition, it lifts the ambiguity on how the orbital wavefunctions are calculated in section 5."
We have substantially revised the explanations and derivations in section 2.2 to clarify all the issues raised by the referee in this point. See especially the text and new equations between Eqs. (29) and (34). In particular, the normalization factor is now explicitly and properly defined with the new symbol N_P (for what was previously c_P V_0), and the nonorthonormality of the spin kets |psi_b(p)> is mentioned explicitly [just below Eq. (25)]. Corrections have also been made in equations outside section 2.2 as necessary.
d.: "For the derivation of Eq. (31) the completeness of the two-body helicity spin-basis <Eq. (2) given in the report> has to be employed. A property that is not mentioned in the current form of the manuscript. ... To make the derivation more transparent this property should be added in section 2.1 where the two-body helicity basis is first introduced. This comment should be subsequently referred to in the discussion regarding Eq. (31)."
We have added the completeness relation given as Eq. (2) in the report as new Eq. (19) in section 2.1, introduced by the sentence ‘For fixed P ... this subspace;’. As suggested by the referee, this relation is then referred to in the discussion just above Eq. (41) [the previous Eq. (31)] ‘Employing the resolution (19) ... ‘.
Minor comments:
1.: "In Eq. (2) h is introduced but is not defined as denoting the Zeeman energy. The authors should make sure that this quantity is defined in the updated version of their manuscript. In addition, they might consider changing the notation from h to Delta to avoid any possible confusion with the Planck constant."
We now define the Zeeman energy h explicitly after Eq. (2). We decided to stick with the symbol h, as it is widely adopted in the related literature and to avoid a possible confusion of the symbol Delta with an energy gap or pair potential, e.g., in a Bogoliubov-deGennes Hamiltonian.
2.: "In the discussion following Eq. (3), it might be useful to note that M provides a transformation from the configuration to spin space. Moreover, the authors might consider amending the expression regarding lambda(p) by including the expansion of the matrix multiplications <mathematical expression in report>. This change will make the distinction between spatial coordinates and spin coordinates clear."
We found this remark really useful and have included its message in the new paragraph above Eq. (4) [previous Eq. (3)]. In particular, we added the mathematical expression suggested by the referee as our new Eq. (3), and we now refer to q as the ‘vector of momentum-dependent spin splittings’ to distinguish it from the real-space orbital vector p.
3.: "Regarding Eq. (10) it should be commented that alpha_1 and alpha_2 take values from -1 to 1 and describe the helicity states."
We have added the requested information below Eq. (14) [previously Eq. (10)].
4.: "In Eq. (23) it might be preferable to denote c_P as a function c(P), or even as N(P), where the latter points out its role as the normalization constant."
As part of our revisions in section 2.2, we have chosen to absorb c_P into the new normalization factor N_P. See, e.g., Eq. (31) [previously Eq. (22)].
5.: "The term ‘secular equation’ might be unfamiliar to some physicists, it might be a good idea to replace it with some more familiar term characteristic or eigenvalue equation."
We have replaced ‘secular’ by ‘characteristic’.
6.: "In Eq. (29) the transformation from momentum to real space is only done for the relative coordinate while keeping the COM momentum constant, I think that also the latter fact should be mentioned."
We have added a footnote (footnote 1) below Eq. (39) [previously Eq. (29)] to clarify this point.
7.: "During the discussion regarding the reduction to dimensionless quantities, it should be noted that this procedure results in the various quantities mentioned thereafter being measured in the characteristic units of the spin-orbit coupling strength, i.e. ... ."
We have added the passage ‘Introducing characteristic units ... define the dimensionless quantities’ below Eq. (46) to convey the suggested information. We also decided to display the definitions of the rescaled energies more explicitly in the new Eqs. (47).
8.: "When the dimensionless threshold energy \tilde{E}_th is introduced in section 3.1, Eq. (15), which defines this quantity, should also be mentioned as a reminder."
In the revised manuscript, an extended discussion of the energy threshold is now provided at the beginning of section 2.2, which is referenced when introducing \tilde{E}_th in section 3.1 [below Eq. (49)].
9.: "Additionally, at the beginning of section 3.2 the ‘quantity appearing on the r.h.s. of Eq. (37)’ should be replaced by the concrete specification of the quantity, namely, F_P(epsilon_0, h) for P<>0."
We have made that replacement as recommended.
10.: "The quantity \tilde{P} is undefined within the main text, with its definition given only in the caption of Fig. 3. The authors should also define it where it first appears in the main text."
In the revised manuscript, \tilde{P} is now introduced already in the second paragraph of section 3.2.
11.: "During the discussion in section 4, it remains unresolved how the P_y component of the COM affects the results and one has to trace back to Eq. (8) and (9) to verify that it contributes to a ... shift of the bound state energy. The authors should briefly comment on this in section 4."
We have added the sentence ‘The other COM momentum component ... [see Eq. (21)].’ in the first paragraph of section 4 to point out where P_y really matters.
12.: “In the end of section 4, to emphasize the fact that ‘This is important for experiments, as 1D-type spin-orbit coupling is easier to realize, but tuning the interactions to a sufficiently weak strength (e.g. near the zero-crossing of a Feshbach resonance) might prove challenging.’, it would be important to include the appropriate citations.”
We have reworded the corresponding sentence as the revised results discussed below (and seen in the new Fig. 6b) imply less stringent requirements for the interaction strength to be weak. Instead, we point to the potential experimental challenges in tuning to a narrow parameter regime where the binding energy is small and point to Ref. [1] as a general reference to the experimental requirements for 1D-type spin-orbit coupling. This change affects the last sentence of Sec. 4 starting with “While 1D-type spin-orbit coupling …”.
13.: A brief discussion about the envisaged experimental probes should be included in the introduction of the manuscript.
We have added the sentence ‘The polarized spin-triplet character …’ discussing experimental probes to the second-to-last paragraph of the introduction, as suggested.
- In section 6, which describes the possible experimental detection, a criterion referring to a characteristic momentum scale ... is derived. The authors should briefly comment on how this momentum scale compares to the resolution that is experimentally achievable considering the typical temperatures in state-of-the-art ultracold atom experiments.
The issue of experimental resolution has been addressed in the slightly rewritten section 6, with reference to the experimental work cited as Ref. [44]. Specifically, the changes affect the new sentences ‘As the square root … experiments’ after the formula for the momentum scale and ‘Due to the single-particle resolution … post selection’.
Response to Report 2 and associated changes:
We thank the Referee for their very supportive report. The requested changes/clarifications have been implemented in the revised manuscript as follows:
1.: “The parameter b defined below Eq. (5) gives the relative strength of the Zeeman and SO interaction seems to have the dimension of momentum. Does this parameter then adequately reflects the relative strength of these Zeeman and SO interactions. Would it not be better to use a dimensionless ratio of interaction strengths?”
The appropriate dimensionless parameter measuring the relative strengths of Zeeman splitting and spin-orbit coupling is \tilde{h}=h/(m lambda^2) defined in new Eq. (47b) in the revised manuscript. In contrast, the quantity b=h/lambda corresponds to the momentum scale where Zeeman splitting and spin-orbit spin splitting coincide. After pondering the Referee’s question, we decided that it would be best to eliminate b from the formalism so as to avoid any confusion about its meaning. Hence, we redefined the quantity Z(p) as in new Eq. (6) and made all necessary adjustments in formulae. Now h appears wherever b used to be, which makes physical interpretations much clearer.
2.: “Equations (9) establish that the relative motion of two particles in the COM frame depends on the COM momentum P. This is an unusual situation because the reason one introduces the COM and relative coordinates in the first place is to decouple the Hamiltonian into two commuting parts (the COM and internal Hamiltonians). Hence, two points should be clarified. First, is there really any practical advantage to using the coordinates in Eq. (6)? Second, it would be helpful to mention exactly which interactions are responsible for the coupling between the external and internal degrees of freedom. Are these the SO coupling terms of the kind \sigma_x p_y?”
Yes, the transformation to relative and COM coordinates is helpful because it reduces the dimensionality of the problem from four degrees of freedom (two particles in two spatial dimensions) to two separate problems with two degrees of freedom each (the COM motion being that of a free particle). While the COM momentum still enters the relative-motion problem, it does so as parameter only, which can be considered to have a fixed value. We have added a discussion of the benefits of COM-motion separation in the last paragraph before the start of Sec. 2.1 (‘While the coordinate … generate such terms.’) where we also explain that all forms of spin-orbit coupling considered in this work generate the COM-momentum-dependent terms in the relative-motion problem.
3.: “In Section 2.2 the authors use the Green function approach to obtain the bound states of s-wave interacting particles. Is this the only approach that can be used? It would also be helpful for the general reader to understand how this approach works using a simple example. A reference to the approach being applied to, e.g., two interacting particles in the absence of the SO interaction, would be helpful.
We have added a new paragraph at the beginning of Sec. 2.2 to point the reader to the relevant previous studies of 2D-fermion bound states upon which our work builds, including the seminal work on pairing in 2D without spin-orbit coupling referred to in new Ref. [47].
Additional changes made in the resubmitted manuscript:
We felt it would be useful to better describe the origin and value of the energy threshold below which stable bound states exist. We also realized that our previous application of the energy threshold was incorrect in some parameter regimes. To this end, we have made the following changes:
- new Eq. (8) gives the minimum of the single-particle energy dispersion,
- new Eq. (18) gives the minimum of the relative-motion contribution to the two-particle energy, and
- the second paragraph of section 2.2 now contains an extensive discussion about energy thresholds.
In particular, we now explain that two different energy thresholds need to be distinguished, an absolute one [E_th^abs, given in new Eq. (21)] that constitutes the absolute minimum of energy available to two unbound particles, and the threshold E_th^rel given in new Eq. (22) that is the minimum of energy available to two unbound particles with their COM momentum fixed at a given value. We elucidate the relationship between these two thresholds and their different physical meanings to motivate our definition of the bound-state energy using E_th^rel [see new Eq. (23)]. As the two thresholds differ when the COM momentum is finite, we include the critical line corresponding to the more stringent E_th^abs in the plot shown in new Fig. 4(a).
In the process of reconsidering the threshold energy, we realized that our previous definition for E_th does not apply for \tilde{h}>1. As a result of using the wrong E_th, we had excluded a region in parameter space where a stable bound state also exists; this is the region between the dashed black and the solid blue curves in the new Fig. 1(a). We rectified this error, derived the new critical boundary given in Eq. (51) and illustrate its form in the new Fig. 2. We furthermore recalculated the triplet-state admixture for the extended region in parameter space where a bound state exists and find that also the region of triplet-dominated dimers is enlarged. See new Fig. 3. Thus, the new results strengthen our conclusions regarding triplet character of bound states and extend the region of experimental accessibility. We have also recalculated Figs. 4, 5 and 6 (previously Figs. 3, 4 and 5) to implement the correct threshold for the other situations considered in our article (finite COM momentum or 1D-type spin-orbit coupling). The discussion of these results in the main text has been adjusted as needed. All main conclusions remain the same, with the only change being the overall greater prevalence of triplet-dominated dimers. Figures 7, 8 and 9 (previously Figs. 6, 7 and 8) remain unchanged.
The discussion of mathematical details presented in Appendices C and D has been adjusted and extended to account for the corrected form of the energy threshold.
The new references [50] and [51] have been included when discussing features of the new critical boundary Eq. (51), in particular, how it differs from the two-particle version of the Chandrasekhar-Clogston criterion.

---

## Round 2 · List of Changes

Extensive changes were made to the text and figures in response to the referees’ comments and due to the need of distinguishing the two separate dissociation thresholds (where some the previous formulas and figures were incorrect), as detailed in the authors’ response section.

---

## Editorial Decision

published